# SaNN: Simple Yet Powerful Simplicial-aware Neural Networks

**Sravanthi Gurugubelli & Sundeep Prabhakar Chepuri**
Indian Institute of Science, Bangalore, Karnataka, India
{sravanthig,spchepuri}@iisc.ac.in

## Abstract

Simplicial neural networks (SNNs) are deep models for higher-order graph representation learning. SNNs learn low-dimensional embeddings of simplices in a simplicial complex by aggregating features of their respective upper, lower, boundary, and coboundary adjacent simplices. The aggregation in SNNs is carried out during training. Since the number of simplices of various orders in a simplicial complex is significantly large, the memory and training-time requirement in SNNs is enormous. In this work, we propose a scalable simplicial-aware neural network (SaNN) model with a constant run-time and memory requirements independent of the size of the simplicial complex and the density of interactions in it. SaNN is based on pre-aggregated simplicial-aware features as inputs to a neural network, so it has a strong simplicial-structural inductive bias. We provide theoretical conditions under which SaNN is provably more powerful than the Weisfeiler-Lehman (WL) graph isomorphism test and as powerful as the simplicial Weisfeiler-Lehman (SWL) test. We also show that SaNN is permutation and orientation equivariant and satisfies simplicial-awareness of the highest order in a simplicial complex. We demonstrate via numerical experiments that despite being computationally economical, the proposed model achieves state-of-the-art performance in predicting trajectories, simplicial closures, and classifying graphs.

## 1 Introduction

Graph Neural Network (GNN) models are extensively used for analyzing graph-structured data by embedding nodes as points in Euclidean space through neighborhood feature aggregation (Hamilton et al., 2017; Leskovec & Jegelka, 2019; Velickovic et al., 2017; Rossi et al., 2020; Chen et al., 2020). The expressive power of a GNN model is often benchmarked against the Weisfeiler-Lehman (WL) isomorphism test (Lehman & Weisfeiler, 1968) and GNN architectures as powerful (in terms of expressiveness) as the WL test can be designed (Leskovec & Jegelka, 2019) by appropriately choosing the aggregation functions. However, an inherent limitation of any graph-based neural model lies in its ability to encode only pairwise interactions between entities. In many real-world scenarios, interactions transcend pairwise relationships, for instance, group-based interactions are seen in biochemistry (e.g., reaction between reagents), social networks (e.g., interactions between friends), and trade networks (e.g., interaction between buyers, suppliers, and intermediaries), to name a few. Such supra-pairwise interactions can be effectively captured using simplicial complexes, a higher-order generalization of graphs.

A simplicial complex consists of simplices of various orders, including 0-simplices (or nodes), 1-simplices (or edges), 2-simplices (or triangles), and more generally, $k$-simplices (or simplices of order $k$). Higher-order simplices may also be oriented, ensuring a consistent node arrangement within each simplex, facilitating tasks like determining information flow directions along the simplices. In simplicial complexes, $k$-simplices have four types of adjacent simplices: boundary-adjacent ($(k-1)$-simplices), co-boundary-adjacent ($(k+1)$-simplices), upper-adjacent, and lower-adjacent ($k$-simplices). Similar to how GNNs create meaningful embeddings of nodes via sequential neighborhood aggregation, simplicial neural networks (SNNs) generate embeddings for all the $k$-simplices within a simplicial complex. Unlike GNNs that aggregate attributes only from adjacent nodes, SNNs leverage information from upper, lower, boundary, and co-boundary adjacent simplices of a $k$-simplex, enabling them to capture higher-order interactions and generate more expressive

embeddings compared to GNNs (Bodnar et al., 2021). While the expressive power of GNNs is evaluated through the WL test, Bodnar et al. (2021) introduces a theoretical characterization framework for SNNs, namely, the simplicial Weisfeiler-Lehman (SWL) test, which is provably more powerful than the WL test. Further, it is shown that under some conditions on the aggregation functions, SNNs are strictly more powerful than the WL test and are as powerful as the SWL test (Bodnar et al., 2021). Certain SNN models, such as Bodnar et al. (2021) and Roddenberry et al. (2021), also showcase properties like simplicial awareness [1], permutation equivariance [2], and orientation equivariance [3].

An important limitation of current SNN models (Ebli et al., 2020; Bunch et al., 2020; Bodnar et al., 2021; Roddenberry et al., 2021; Yang et al., 2022b), is the necessity of sequential feature aggregation during training, and this incurs significant memory and training time requirements. In this paper, we propose *simplicial-aware neural networks* (SaNN), a simpler model with a strong simplicial inductive bias and a constant training time and memory requirement (independent of the number of interacting simplices) by augmenting pre-aggregated features as inputs to a neural model, such as a multi-layer perceptron (MLP).

**Contributions:** The main contributions and results are summarized as follows:

- We devise a recursive aggregating method with no learnable parameters to compute features of simplices that are aware of their different hops of neighborhood. The proposed precomputation of features results in a training time that is almost constant for simplicial complexes of any size and order.

- We then prescribe conditions on the aggregating functions involved in generating the simplicial-aware features in SaNN to ensure that the embeddings generated by SaNN are provably as expressive as those generated by the message passing simplicial network (MPSN). We theoretically prove that SaNN, although a simpler model, is strictly more powerful than the WL test and as powerful as MPSN, or equivalently, the SWL test (Bodnar et al., 2021).

- We also theoretically show that SaNN is permutation equivariant, orientation equivariant, and satisfies simplicial awareness of the highest order in a simplicial complex while being computationally efficient.

We demonstrate the efficacy of SaNN for three applications: trajectory prediction, simplicial closure (i.e., higher-order link) prediction, and graph classification. We observe that for all the three applications, SaNN outperforms non-deep baselines while being competitive with existing SNN models with a much smaller run-time. Specifically, for simplicial closure prediction, on large datasets with about a million simplices of various orders, existing SNN models run out of memory.

We next discuss a few works that are related to the proposed work.

**Related Works:** While exisiting SNN models (Bodnar et al., 2021; Ebli et al., 2020; Roddenberry et al., 2021; Bunch et al., 2020) differ in how the features of neighborhood simplices are aggregated and how the aggregated features from different types of neighborhoods are combined, Yang et al. (2022b) addresses scalability issues in higher-order graph-based neural networks. Unlike existing SNN models, it relaxes the fundamental definition of simplicial complexes, namely, the inclusion principle (subsets of simplices are simplices in the simplcial complex), to reduce runtime complexity. For tasks like predicting simplicial closure, the inclusion principle cannot be ignored Benson et al. (2018). For instance, in predicting whether a simplex is likely to form between three authors who have not all collaborated before, Benson et al. (2018) demonstrates that the frequency of collaboration between any two authors in the group positively affects the closure of the triangle. Moreover, Yang et al. (2022b) assumes all simplices in a complex are unoriented, rendering it unsuitable for tasks dependent on relative orientation, such as trajectory prediction (Roddenberry et al., 2021).

---

[1]Simplicial-awareness ensures the dependence of embeddings on the simplices of all orders within a simplicial complex.

[2]Permutation equivariance means that the embeddings of simplices remain unaltered even as the simplices are reordered, an important attribute for modeling complex structures like graphs and simplicial complexes.

[3]Orientation equivariance implies that changing the relative orientations of certain simplices will only lead to the output embeddings of those simplices being the sign-inverted versions of the embeddings prior to the orientation change.

The proposed work is inspired by graph-based neural network models that attempt to simplify GNN models to improve their scalability. Graph-augmented multi-layer perceptrons (GAMLPs) (Chen et al., 2020), simple and parallel graph isomorphism network (SPIN) (Doshi & Chepuri, 2022), scalable inception graph neural networks (SIGN) (Rossi et al., 2020), and simple graph convolutional networks (SGCNs) (Wu et al., 2019) are some examples of scalable and efficient GNN models. GAMLPs, SPIN, and SIGN are related to `SaNN` in that they compute the node features as a preprocessing step before the training procedure. However, they are limited to generating node embeddings by accounting only the pairwise relations. Furthermore, the notion of a neighborhood in `SaNN` differs from that in GAMLPs, SPIN, or SIGN. A direct extension of the precomputation step in GAMLPs, SPIN, or SIGN to simplicial complexes would be to precompute the features of simplices using the integer powers of the so-called Hodge Laplacian matrix (the generalization of the graph Laplacian for simplicial complexes and an aggregation operator used in Ebli et al. (2020); Bunch et al. (2020). However, it does not account for information in the boundary and co-boundary adjacent simplices, which as we prove in this work, is required to propose an efficient model that is as powerful as the SWL test. Specifically, we theoretically prove that for a specific choice of functions involved in generating embeddings, `SaNN` is strictly more powerful than the existing GNN models. In other words, `SaNN` implicitly, is strictly more powerful than GAMLPs, SPIN, or SIGN. This also implies that even the node embeddings (i.e., embeddings of 0-simplices) from SaNN are more expressive than the 1-WL test or, equivalently, GNNs. Using the node embeddings alone, `SaNN` distinguishes a broader class of graphs than those that are distinguishable by GNNs. In summary, our proposed SNN model, denoted by `SaNN`, is significantly faster than existing SNN models while preserving the definition of simplicial complexes and the expressive power of the current SNN models.

## 2 BACKGROUND

In this section, we mathematically describe simplicial complexes and SNNs.

**Simplicial Complex:** Let $\mathcal{V} = \{v_0, v_1, \ldots, v_N\}$ be the node set of cardinality $N + 1$. A simplicial complex $\mathcal{K}$ is a collection of non-empty subsets of $\mathcal{V}$ with the inclusion property that if $\sigma$ is an element of $\mathcal{K}$, then every subset of $\sigma$ is an element of $\mathcal{K}$. A $k$-dimensional element $\sigma = \{v_0, v_1, \ldots, v_k\}$ of $\mathcal{K}$ with cardinality $k + 1$ is called a $k$-simplex. Each simplex has an orientation defined by a standardized vertex order, typically ascending or descending, establishing a standard node arrangement within each simplex. The number of $k$-simplices in $\mathcal{K}$ is denoted by $N_k$. A simplicial complex is said to have an order of $K$ if the cardinality of its largest simplex is $K + 1$. Such a simplicial complex is also referred to as a $K$-simplicial complex.

A $k$-simplex $\sigma_k$ has four kinds of adjacent simplices, namely, boundary adjacent, co-boundary adjacent, upper adjacent, and lower adjacent simplices. The incidence relationship between $(k-1)$-simplices and $k$-simplices along with their relative orientations can be represented by the oriented incidence matrix $\mathbf{B}_k \in \mathbb{R}^{N_{k-1} \times N_k}$. The $(i, j)$th entry of $\mathbf{B}_k$ is non-zero if the $i$th $(k-1)$-simplex is a boundary simplex of the $j$th $k$-simplex. The non-zero entries of an oriented incidence matrix $\mathbf{B}_k$ can be either $+1$ or $-1$, reflecting the relative orientations of the $i$th $(k-1)$-simplex and the $j$th $k$-simplex. For unoriented simplices, we use unoriented incidence matrices in which the $(i, j)$th entry is 1 if the $i$th $(k-1)$-simplex is a boundary simplex of the $j$th $k$-simplex, and 0 otherwise. The upper and lower adjacencies of $k$-simplices can be defined using the upper and lower Laplacian matrices as $\mathbf{A}_{k,\mathcal{U}} = \mathbf{B}_{k+1}\mathbf{B}_{k+1}^{\mathrm{T}} \in \mathbb{R}^{N_k \times N_k}$ and $\mathbf{A}_{k,\mathcal{L}} = \mathbf{B}_k^{\mathrm{T}}\mathbf{B}_k \in \mathbb{R}^{N_k \times N_k}$, respectively. We will also define the matrices $\mathbf{A}_{k,\mathcal{B}} = \mathbf{B}_{k+1}^{\mathrm{T}} \in \mathbb{R}^{N_{k+1} \times N_k}$ and $\mathbf{A}_{k,\mathcal{C}} = \mathbf{B}_k \in \mathbb{R}^{N_{k-1} \times N_k}$, for convenience. The $t$-hop upper, lower, boundary, and co-boundary neighbors of a simplex $\sigma_k$ are denoted by $\mathcal{U}^{(t)}(\sigma_k)$, $\mathcal{L}^{(t)}(\sigma_k)$, $\mathcal{B}^{(t)}(\sigma_k)$, and $\mathcal{C}^{(t)}(\sigma_k)$, respectively. For example, $\mathcal{U}^{(1)}(\sigma_0)$ for a 0-simplex $\sigma_0$ simply means (upper adjacent) neighborhood of the node $\sigma_0$.

**Simplicial Neural Networks:** Let us denote the attributes of $k$-simplices by $\mathbf{X}_k \in \mathbb{R}^{N_k \times D_k}$ and the $D_k$-dimensional feature of a $k$-simplex $\sigma_k$ as $\mathbf{X}_k[\sigma_k]$. The sign of feature $\mathbf{X}_k[\sigma_k]$ is determined based on the reference orientation of $\sigma_k$. For unoriented features, there is no need for a reference orientation of simplices. In such cases, we consider unoriented incidence matrices. SNNs take the simplicial complex structure and the data matrices $\mathbf{X}_k$ for $k = 0, \ldots, K$ as input, and update the embeddings of the simplices by aggregating embeddings of their adjacent simplices. A generic way of expressing the update rule of the embeddings of $k$-simplices by the existing SNN models in the $l$th layer is given by

$$\mathbf{H}_k^{(l+1)} = \phi\left[\psi\left(\mathbf{A}_{k,\mathcal{S}}\mathbf{H}_k^{(l)}\mathbf{W}_{\mathcal{S}}^{(l)}, \mathbf{A}_{k,\mathcal{U}}\mathbf{H}_k^{(l)}\mathbf{W}_{\mathcal{U}}^{(l)}, \mathbf{A}_{k,\mathcal{L}}\mathbf{H}_k^{(l)}\mathbf{W}_{\mathcal{L}}^{(l)}, \mathbf{A}_{k-1,\mathcal{B}}\mathbf{H}_{k-1}^{(l)}\mathbf{W}_{\mathcal{B}}^{(l)}, \mathbf{A}_{k+1,\mathcal{C}}\mathbf{H}_{k+1}^{(l)}\mathbf{W}_{\mathcal{C}}^{(l)}\right)\right], \quad (1)$$

where $\phi$ is a non-linear function (e.g., ReLU or sigmoid), $\psi$ is a combining function (e.g., summation or concatenation) that combines information from the different neighborhoods, and $\mathbf{H}_k^{(0)} = \mathbf{X}_k$. The matrix $\mathbf{A}_{k,\mathcal{S}}$ lets the model include the self-embeddings of simplices while updating their respective embeddings. Specifically, the matrix $\mathbf{A}_{k,\mathcal{S}}$ is taken as the identity matrix $\mathbf{I}$ if self-embeddings are to be accounted for, and is otherwise set to an all-zero matrix $\mathbf{0}$. The matrices $\{\mathbf{W}_{\mathcal{S}}^{(l)}, \mathbf{W}_{\mathcal{U}}^{(l)}, \mathbf{W}_{\mathcal{L}}^{(l)}, \mathbf{W}_{\mathcal{B}}^{(l)}, \mathbf{W}_{\mathcal{C}}^{(l)}\}$ are learnable weight matrices of the $l$th layer. A model with $L$ such layers in cascade sequentially learns embeddings of $k$-simplices using information from their $L$-hop neighborhood.

Different choices of the functions, $\phi$ and $\psi$, along with aggregating matrices result in various SNN models, namely, MPSN Bodnar et al. (2021), SCoNE Roddenberry et al. (2021), SCNN Yang et al. (2022a), S2CNN Bunch et al. (2020), and SNN Ebli et al. (2020). More details about the specific choice of the functions in each of the SNN models is provided in Appendix A.

## 3 THE PROPOSED MODEL

The key idea behind the proposed model is to precompute simplicial structure-aware features by aggregating initial features of simplices of different orders from different neighborhood hops. The aggregated features are then transformed using nonlinear functions to obtain embeddings for simplices of different orders. The generic embeddings can be used to obtain task-specific embeddings for the desired application. The proposed generic SaNN model for each of the $k$-simplices has the following two main components.

**Precomputing Simplicial-aware Features:** Let us collect the feature vectors in the $t$-hop upper neighborhood of a $k$-simplex $\sigma_k$ in the multiset[4] $\mathbb{X}_{k,\mathcal{U}}^{(t)}[\sigma_k] = \left\{\!\!\left\{ \mathbf{X}_k[\tau], \forall \tau \in \mathcal{U}^{(t)}(\sigma_k) \right\}\!\!\right\}$. Similarly, define the following multisets $\mathbb{X}_{k,\mathcal{L}}^{(t)}[\sigma_k] = \left\{\!\!\left\{ \mathbf{X}_k[\tau], \forall \tau \in \mathcal{L}^{(t)}(\sigma_k) \right\}\!\!\right\}$, $\mathbb{X}_{k,\mathcal{B}}^{(t)}[\sigma_k] = \left\{\!\!\left\{ \mathbf{X}_{k-1}[\tau], \forall \tau \in \mathcal{B}^{(t)}(\sigma_k) \right\}\!\!\right\}$, and $\mathbb{X}_{k,\mathcal{C}}^{(t)}[\sigma_k] = \left\{\!\!\left\{ \mathbf{X}_{k+1}[\tau], \forall \tau \in \mathcal{C}^{(t)}(\sigma_k) \right\}\!\!\right\}$. We sequentially pre-compute simplicial-aware features from different neighborhood depths as follows. We update the feature vector of a $k$-simplex $\sigma_k$ by aggregating $t$-hop information of $\sigma_k$, using the following partial updates that are dependent on $(t-1)$-hop aware features as

$$\mathbf{Y}_{k,\mathcal{U}}^{(t)}[\sigma_k] = f_{k,\mathcal{U}}(\mathbb{X}_{k,\mathcal{U}}^{(t-1)}[\sigma_k]), \mathbf{Y}_{k,\mathcal{L}}^{(t)}[\sigma_k] = f_{k,\mathcal{L}}(\mathbb{X}_{k,\mathcal{L}}^{(t-1)}[\sigma_k]),$$
$$\mathbf{Y}_{k,\mathcal{B}}^{(t)}[\sigma_k] = f_{k,\mathcal{B}}(\mathbb{X}_{k,\mathcal{B}}^{(t-1)}[\sigma_k]), \mathbf{Y}_{k,\mathcal{C}}^{(t)}[\sigma_k] = f_{k,\mathcal{C}}(\mathbb{X}_{k,\mathcal{C}}^{(t-1)}[\sigma_k]), \quad (2)$$

Figure 1: Simplicial-aware neighborhood aggregation involves iterative aggregation (no learnable parameters) from the upper, lower, boundary, and co-boundary simplices of $k$-simplices.

where $f_{k,n} : \mathbb{X}_{k,n}^{(t)}[\sigma_k] \to \mathbb{R}^{D_k}$, for $n \in \{\mathcal{U}, \mathcal{L}, \mathcal{B}, \mathcal{C}\}$ are the neighborhood aggregation functions that aggregate features of the $k$-order upper, $k$-order lower, $(k-1)$-order boundary, and $(k+1)$-order co-boundary adjacent simplices, respectively, of any $k$-simplex. The final aggregated $t$-hop aware embedding of a $k$-simplex $\sigma$ is computed by combining the partial updates obtained from the $(t-1)$-hop aware embeddings as

$$\mathbf{X}_k^{(t)}[\sigma_k] = \phi\left( \mathbf{Y}_{k,\mathcal{U}}^{(t)}[\sigma_k], \mathbf{Y}_{k,\mathcal{L}}^{(t)}[\sigma_k], \mathbf{Y}_{k,\mathcal{B}}^{(t)}[\sigma_k], \mathbf{Y}_{k,\mathcal{C}}^{(t)}[\sigma_k] \right) \quad (3)$$

for $t = 0, 1, \cdots, T$ with $\mathbf{X}_k^{(0)} = \mathbf{X}_k$ as the initial features, where $\phi$ is a function that combines aggregated features from the four types of neighborhoods and $T$ denotes the depth of neighborhood information considered by SaNN (which is analogous to the number of layers in SNNs). For any $k$-simplex $\sigma_k \in \mathcal{K}$, the aggregated feature vectors $\mathbf{X}_k^{(t)}[\sigma_k]$ can be efficiently precomputed for all $k$ and $t$ outside the training process as no learnable parameters are involved, but they have a strong simplicial-structural inductive bias. The aggregation scheme in an SaNN model is illustrated in Fig. 1.

**Learning from Simplicial-aware Features:** We use learnable nonlinear transformation functions, $g_k^{(t)}(\cdot)$, to transform the precomputed features into embeddings that are suitable for the task at hand as

$$\mathbf{S}_k^{(t)} = g_k^{(t)}\left( \mathbf{X}_k^{(t)} \right) \quad (4)$$

for $k = 0, 1, \cdots, K$ and $t = 0, 1, 2, \cdots, T$. We finally combine the embeddings of $k$-simplices from different hops of neighborhood as

$$\mathbf{H}_k^{(t)} = \Theta_k\left( \mathbf{S}_k^{(0)}, \mathbf{S}_k^{(1)}, \cdots, \mathbf{S}_k^{(t)} \right), \quad (5)$$

Figure 2: Feature transformation blocks of SaNN compute the matrix embedding $\mathbf{H}_k$ of $k$-simplices from the precomputed features. Here, we use plate notation, with $K+1$ at the bottom of the plate denoting the presence of $K+1$ such blocks for $k = 0, \ldots, K$.

---

[4]A multiset is a collection of elements that accommodates duplicate elements while accounting for the multiplicities of each element.

where $\Theta_k$ is a function that combines information from different hops for $k$-simplices. The transformation of precomputed features and the combination of features from different depths in an SaNN model is summarized in Fig. 2. We denote the final embedding of a $k$-simplex $\sigma_k \in \mathcal{K}$ generated by SaNN using the information from $0, 1, \ldots, T$-hop neighborhood as $\mathbf{H}_k^{(T)}[\sigma_k] =: \mathbf{H}_k[\sigma_k]$.

**Computational Complexity:** SaNN model incurs significantly lower time complexity than the existing SNN models as we are precomputing the aggregated features from different hops while allowing the features to have simplicial-structural inductive bias. Specifically, the existing $T$-layer SNN models have an overall time complexity of about $\mathcal{O}(T((2N_k^2 D_k + N_k N_{k-1} D_{k-1} + N_k N_{k+1} D_{k+1}) + (3N_k D_k^2 + N_k D_{k-1}^2 + N_k D_{k+1}^2)))$, while an SaNN model (an example architecture is provided in Section 4.1) capturing information from $0, \ldots, T$-hop neighborhood of $k$-simplices has a significantly smaller time complexity of $\mathcal{O}(T(3N_k D_k^2 + N_k D_{k-1}^2 + N_k D_{k+1}^2))$ compared to the existing SNN models. More details on the contribution of different components of SNNs and SaNN to their respective computational complexities are provided in Appendix B. In Fig. 3, we show a comparison of the average run-time measurements of SaNN and MPSN on an example dataset. We observe that the average run-time of the proposed

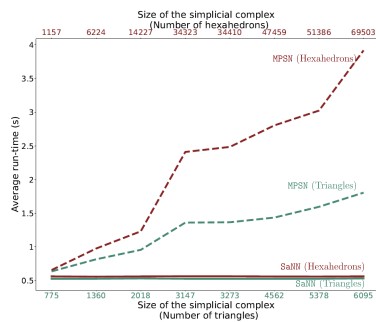

Figure 3: Average run-time of SaNN and MPSN (over 20 experiments of 50 epochs) on email-Enron dataset (Benson et al., 2018), with triangles and hexahedrons as highest order simplices.

model for a forward pass is almost constant for simplicial complexes of any size and increases only slightly with simplices of higher orders, whereas the run-time of MPSN increases drastically when simplices of higher orders are considered.

In what follows, we theoretically characterize the expressive power of SaNN.

## 4 THEORETICAL CHARACTERIZATION OF SaNN

Given the computational advantage of the SaNN model, we now analyze its expressive power. Specifically, we characterize the discriminative power of SaNN with respect to the WL and SWL tests (Please refer to Appendix C for more details about the SWL test). The following theorem states the conditions under which the SaNN model is more powerful than the WL test.

**Theorem 4.1.** *SaNN is strictly more powerful than the WL test in distinguishing non-isomorphic graphs with a complex-clique lifting if all the functions involved in generating the node embeddings, namely, $f_{0,\mathcal{U}}(\cdot), f_{0,\mathcal{C}}(\cdot), \phi(\cdot), g_0^{(t)}(\cdot),$ and $\Theta_0(\cdot)$ for $t = 0, \ldots, T$ are injective.*

Although MPSN is provably more powerful than the WL test, it has the same form of the sequential approach of aggregating transformed features as is the case with the WL test. The proposed model, however, does not have the same form as the WL test since it is based on pre-aggregating features from different hops. Hence, the proof of comparing the expressive powers of SaNN and the WL test is not trivial. The proof of Theorem 4.1 is provided in Appendix D. To prove the above theorem, we first prove that SaNN is atleast as powerful as the WL test by showing that if the node embeddings of two nodes generated by SaNN are equal, then the WL node coloring for the two nodes are equal. We then give an example where SaNN, using the higher-order information, distinguishes two graphs that the WL test cannot, to show that SaNN is more powerful than the WL test. Thus for appropriately chosen functions in Equations (3), (4), and (5), SaNN is provably more powerful than the WL test.

The theorem implies that any arbitrary extension of GAMLPs (Chen et al., 2020), SPIN (Doshi & Chepuri, 2022), or SIGN (Rossi et al., 2020) to higher-order simplices does not result in the node-embeddings from SaNN having a superior expressive power GNNs. In one possible extension, we could replace the integer powers of adjacency matrices in these graph models with those of the Hodge Laplacian matrices. These matrices generalize the graph Laplacian to simplicial complexes and are defined as the sum of upper and lower Laplacians. However, even with this modification, the expressive power of these extended models does not surpass that of GNNs. The theorem states that the node embeddings from the proposed model, under the conditions given in Theorem 4.1, are more expressive than those from the WL test (or equivalently, any of the existing GNN models and scalable versions). Considering a graph with no higher-order interactions, the node embeddings (0-simplices) from SaNN have better expressive power than scalable GNNs because SaNN incorporates information from its co-boundary adjacent simplices (edges) while learning the embedding for a node, which inturn carry information from their co-boundary adjacent simplices (triangles), and so on.

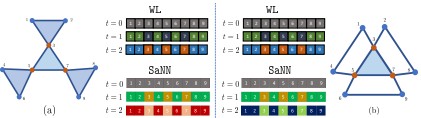

Figure 4: A pair of non-isomorphic graphs (a) and (b) distinguishable by SaNN but not by the WL test. The histograms of colors assigned by the WL test and those of embeddings assigned by SaNN for the first three iterations are displayed alongside the two graphs. Here, nodes are assigned numerical labels to facilitate convenient referencing. While the histogram of colors assigned by the WL test to both graphs remains the same, the figure demonstrates that SaNN generates different embeddings.

An illustration to visualize the expressive power of SaNN in comparision to the WL test is provided in Fig. 4. We present one instance of a pair of clique-lifted graphs for which the WL test assigns the same representations and, therefore, fails to identify the two graphs as being non-isomorphic while SaNN assigns different representations to the two graphs and distinguishes them. To assign a representation for the graph as a whole, we follow the usual procedure of constructing the histogram of representations of the nodes. The histogram of colors assigned by the WL test and the histogram of embeddings assigned by SaNN for the first three iterations are shown next to the two graphs. The histogram of colors assigned by the WL test to both graphs is the same. However, as shown in the figure, SaNN generates different embeddings to the two graphs (more details in Section 4).

In the next theorem, we provide the conditions on the functions such that SaNN is as powerful as the SWL test.

**Theorem 4.2.** *SaNN is as powerful as the SWL test in distinguishing non-isomorphic simplicial complexes if the functions involved in generating the embeddings of the simplices, namely, $f_{k,\mathcal{U}}(\cdot), f_{k,\mathcal{L}}(\cdot), f_{k,\mathcal{B}}(\cdot), f_{k,\mathcal{C}}(\cdot), \phi(\cdot), g_k^{(t)}(\cdot),$ and $\Theta_k(\cdot)$ for $t = 0, \ldots, T$ and $k = 0, \ldots, K$, where $T$ is the depth of the neighborhood information considered by SaNN in generating embeddings of simplices of each of the $K + 1$ orders, are injective.*

We prove Theorem 4.2 in Appendix E. To prove the above theorem, we propose a simpler yet equally expressive alternative representation of the SWL update. Using the alternative update rule, we first prove that SaNN is at most as powerful as the SWL test by showing that if the colors assigned by SWL are the same for two simplices, then SaNN also generates the same embeddings for the two simplices. We then prove that SaNN is at least as powerful as the SWL test by showing that if the embeddings generated by SaNN are the same for two simplices, then SWL also generates the same colors for the two simplices.

The theorem states that the embeddings of simplices from the proposed computationally efficient method, under the conditions given in Theorem 4.2, are as expressive as those from Bodnar et al. (2021), which is proved to be as powerful as the SWL test. While the SWL test and Bodnar et al. (2021) are both based on the sequential approach of aggregating transformed features, the pre-aggregated features are transformed only during training in the proposed method. Despite avoiding the non-linear transformation of features in every iteration of feature aggregation, we prove that SaNN is as powerful as the SWL test.

To conclude, it is sufficient to limit the choice of aggregator and transformation functions to those recommended by Theorems 4.1 and 4.2 to design an SaNN model that is guaranteed to be more powerful than the WL test and as powerful as the SWL test, respectively. In what follows, we discuss a few such choices of the aggregator and transformation functions.

## 4.1 AN EXAMPLE SaNN ARCHITECTURE

In this section, we discuss example functions that fulfill the conditions outlined in Theorem 4 and demonstrate that SaNN is as powerful as the SWL test. The SWL test [cf. Appendix C] distinguishes non-isomorphic simplicial complexes based on structure, assuming uniform initial features (colors) across all simplices. To establish equivalence with the SWL test, we consider simplicial complexes with a uniform scalar feature $a$ on all simplices, without attributing any orientation to the features or the simplices. However, it is worth noting that SaNN can also process oriented features in practice, and in such cases, we work with oriented simplicial complexes (or incidence matrices) as defined in Section 2.

**Precomputing Simplicial-aware Features:** Consider simplicial complexes with the same scalar feature $a$ as initial feature on all the simplices. For such a scenario, one choice of the aggregation functions $f_{k,\mathcal{U}}(\cdot), f_{k,\mathcal{L}}(\cdot),$ $f_{k,\mathcal{B}}(\cdot),$ and $f_{k,\mathcal{C}}(\cdot)$ that preserves injectivity is the summation function. The summation of embeddings of upper, lower, boundary, and coboundary adjacent simplices can be computed efficiently using the (sparse) aggregation matrices defined in Section 2. Specifically, we compute the partial updates in (2) for aggregation based on summation recursively as

$$\mathbf{Y}_{k,\mathcal{U}}^{(t)} = \mathbf{A}_{k,\mathcal{U}}\mathbf{X}_k^{(t-1)}, \ \mathbf{Y}_{k,\mathcal{L}}^{(t)} = \mathbf{A}_{k,\mathcal{L}}\mathbf{X}_k^{(t-1)}, \ \mathbf{Y}_{k,\mathcal{B}}^{(t)} = \mathbf{A}_{k-1,\mathcal{B}}\mathbf{X}_{k-1}^{(t-1)}, \ \mathbf{Y}_{k,\mathcal{C}}^{(t)} = \mathbf{A}_{k+1,\mathcal{C}}\mathbf{X}_{k+1}^{(t-1)}. \quad (6)$$

Another common choice for neighborhood aggregation in GNNs and SNNs is degree-based weighted summation. To implement degree-based weighted summation of neighboring embeddings, we use the following normalized incidence matrices: $\bar{\mathbf{B}}_{k,\mathcal{U}} = \mathbf{D}_k^{-1/2}\mathbf{A}_{k,\mathcal{U}}\mathbf{D}_k^{-1/2}, \ \bar{\mathbf{B}}_{k,\mathcal{L}} = \mathbf{D}_k^{-1/2}\mathbf{A}_{k,\mathcal{L}}\mathbf{D}_k^{-1/2}, \ \bar{\mathbf{B}}_{k,\mathcal{B}} = \mathbf{D}_{k,k-1}^{-1/2}\mathbf{A}_{k-1,\mathcal{B}}\mathbf{D}_{k-1,k}^{-1/2},$ and $\bar{\mathbf{B}}_{k,\mathcal{C}} = \mathbf{D}_{k,k+1}^{-1/2}\mathbf{A}_{k+1,\mathcal{C}}\mathbf{D}_{k+1,k}^{-1/2}.$ Here, $\mathbf{D}_k \in \mathbb{R}^{N_k \times N_k}$ is a diagonal matrix whose $(i, i)$th entry is the number of $k$-simplices that are upper and lower adjacent neighbors of the $i$th $k$-simplex, $\mathbf{D}_{k,k-1} \in \mathbb{R}^{N_k \times N_k}$ is a diagonal matrix whose $(i, i)$th entry is the number of $k-1$-simplices that are boundary adjacent neighbors of the $i$th $k$-simplex, and $\mathbf{D}_{k,k+1} = \mathbb{R}^{N_k \times N_k}$ is a diagonal matrix whose $(i, i)$th entry is the number of $k + 1$-simplices that are co-boundary adjacent neighbors of the $i$th $k$-simplex.

Using normalized incidence matrices to aggregate features has the advantage of bringing all the features to the same scale, thus providing numerical stability. However, it is not always injective. For example, consider the two simplicial complexes in Fig. 5 with $a$ as initial feature on simplices of all orders.

Although the triangles, denoted by $\sigma_1$ and $\sigma_2$, in the two simplicial complexes, have a different number of lower adjacent triangles in their 1-hop neighborhood, the two triangles will be assigned the same lower neighborhood aggregated partial embedding by the degree-based weighted summation aggregator, which, in this case, is $a$. The summation aggregator, on the other hand, will assign two different lower adjacent neighborhood aggregated partial embeddings, specifically, $3a$ and $2a$, to the two triangles with different neighborhoods. In Appendix F, we give a generalized case where the degree-based weighted sum aggregator is not injective, i.e., where the proposed architecture assigns the same partial updates to $k$-simplices in simplicial complexes with different structures.

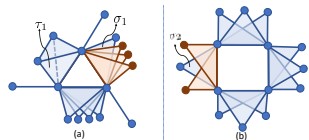

Figure 5: Two simplicial complexes to which SaNN assigns the same partial embeddings of 2-simplices after aggregation from the lower adjacent neighborhood if the degree-based weighted summation aggregator is used.

We get the final updated $t$-hop aware feature vector of a $k$-simplex $\sigma$, denoted by $\mathbf{X}_k^{(t)} \in \mathbb{R}^{N_k \times D_k^{(t)}}$, where $D_k^{(t)} = 2D_k^{(t-1)} + D_{k-1}^{(t-1)} + D_{k+1}^{(t-1)}$, by concatenating the partial updates in (6) as

$$\mathbf{X}_k^{(t)} = \left[\mathbf{Y}_{\mathcal{U}}^{(t-1)}, \mathbf{Y}_{\mathcal{L}}^{(t-1)}, \mathbf{Y}_{\mathcal{B}}^{(t-1)}, \mathbf{Y}_{\mathcal{C}}^{(t-1)}\right]^{\mathrm{T}}, \tag{7}$$

i.e., we use an injective concatenation read-out function (other commonly used sum, mean, or max read-out functions are not injective). Next, we discuss some properties of the proposed aggregation method.

**Property 1.** *The devised aggregation scheme is permutation and orientation equivariant.*

We prove the equivariance of the aggregation method in Appendix G. We first prove the permutation (orientation) equivariance for one-hop aggregation, i.e., we show that if the ordering (orientations) of the input features to SaNN are altered by some permutation operator $\mathcal{P}$, ($\mathcal{O}$, respectively), then the ordering (orientations) of the 1-hop aggregated features are changed by the same operator $\mathcal{P}$, ($\mathcal{O}$, respectively). We then prove, by induction, the permutation (orientation) equivariance of the proposed aggregation for any hop. The equivariance properties of the proposed aggregation method aids in the construction of permutation and orientation equivariant SaNN as we see in the next subsection.

Note that the proposed precomputation method is very different from that in GAMLPs, SPIN, or SIGN. The precomputation step in GAMLPs, SPIN, or SIGN involves taking integer powers of graph adjacency or Laplacian matrices to aggregate information from different hops of neighborhood. However, such a straightforward precomputation step for simplicial complexes will not result in a model that captures information from different types and hops of neighborhood while being (i) as powerful as the SWL test (ii) permutation equivariant, and (iii) orientation equivariant.

**MLPs for Transforming Precomputed Features:** Once the features are aggregated by preserving injectivity and ensuring equivariance properties, the next step is to transform the aggregated simplicial-aware features to more task-aware features using non-linear learnable functions. According to Theorem 4.2, the learnable functions should be injective in order for SaNN to be as powerful as the SWL test. Given the universal approximation ability and injectivity of multi-layer perceptrons (MLPs), we model the transformation functions $g_k^{(t)}$ for $t = 0, \ldots, T$ and $k = 0, \ldots, K$ using MLPs. Single-layer perceptrons, however, are not injective.

For $\Theta_k$ in (5), we use a concatenation read-out function to combine embeddings of simplices of any order $k$ and get the combined embeddings $\mathbf{H}_k$ as

$$\mathbf{H}_k = \left[\mathbf{S}_k^{(0)}, \mathbf{S}_k^{(1)}, \ldots, \mathbf{S}_k^{(T)}\right]. \tag{8}$$

Concatenating the embeddings results in an injective combination of embeddings of simplices of different orders. Typically, to avoid over smoothing, we use embeddings from only 2 or 3-hop neighborhood, thus concatenation does not result in very large dimensional embeddings.

## 4.2 PROPERTIES OF SANN

We next prove that SaNN, with the aggregation and transformation functions satisfying the requirements in Theorem 4.2 (as in the example architecture in the Subsection 4.1), possesses all the equivariance properties that some of the existing SNNs, namely, Roddenberry et al. (2021) and Bodnar et al. (2021), possess. SaNN with the summation-based neighborhood aggregation, feature transformation using MLPs, and using concatenation $\Theta_k$ has the following properties.

**Permutation equivariance:** As discussed in Property 1, the proposed aggregation scheme is permutation equivariant. With the transformation functions as MLPs and the functions that combine embeddings from different hops as concatenation, SaNN is a composition of permutation equivariant functions and hence is permutation equivariant.

**Orientation equivariance:** Using orientation equivariance of the proposed aggregation method [cf. Property 1] and the fact that an MLP is orientation equivariant if the activation functions in the MLP are odd, we prove that SaNN is orientation equivariant if all the activation functions involved in the network are odd in Appendix G.

**Simplicial-awareness:** Since the embeddings of SaNN depend on the boundary matrices of all orders, i.e., $\mathbf{B}_k$ for $k \in 1, \ldots, K$, of a $K$-simplicial complex, SaNN is said to satisfy simplicial-awareness of order $K$ for a $K$-simplicial complex.

## 5 EMPIRICAL EVIDENCE

In this section, we empirically evaluate SaNN in terms of its ability to perform the following three tasks: trajectory prediction, simplicial closure prediction, and graph classification[5].

### 5.1 DOWNSTREAM TASKS

The details of the three tasks are discussed next.

**Trajectory Prediction:** Trajectory prediction involves predicting the next node in a sequence formed by a series of nodes connected by edges, with oriented flows on the edges. As the features are oriented, we use oriented incidence matrices for aggregation. We evaluate the trajectory prediction ability of SaNN on four datasets, namely, *Ocean* (Roddenberry et al., 2021), *Synthetic* (Roddenberry et al., 2021), *Planar* (Cordonnier & Loukas, 2018), and *Mesh* (Cordonnier & Loukas, 2018). We compare the trajectory prediction accuracy of SaNN with state-of-the-art SNN variants, namely, SCoNe (Roddenberry et al., 2021) and SCNN (Ebli et al., 2020), and a non-deep projection based method, denoted by Projection, which projects the input edge features onto the Hodge Laplacian kernel (Schaub et al., 2020).

**Simplicial-closure Prediction:** The goal of simplicial closure prediction is to predict the closure of open simplices in a time series of simplicial complex data. We perform this task on the email-Eu, email-Enron, and contact-high-school datasets, as referenced in Benson et al. (2018). Our methodology involves a temporal split of the data for these datasets, using the initial $80\%$ of the data to train the encoder. The remaining $20\%$ of the data is set aside for inference. Given the highly skewed nature of the dataset, we employ the relative area under the precision-recall curve (AUC-PR) as the evaluation metric for model performance.

**Graph Classification:** Graph classification refers to classifying (clique-lifted) graphs as belonging to one of two or more possible known classes. For all graph classification experiments, we consider the initial features on simplices as the cumulative count of their lower and upper adjacent simplices. As these are unoriented, we use unoriented incidence matrices for aggregation. We compare the graph classification accuracy of SaNN with MPSN (Bodnar et al., 2021) and state-of-the-art GNN variants, DGCNN (Phan et al., 2018), GIN (Leskovec & Jegelka, 2019), and GraphSage (Hamilton et al., 2017), on benchmark datasets for binary as well as multi-class graph classification tasks from chemical and social domains. Specifically, we evaluate on the following TUDatasets (Morris et al., 2020): *Proteins*, *NCI1*, *IMDB-B*, *IMDB-M*, *Reddit-B* and *Reddit-M*.

### 5.2 RESULTS AND DISCUSSION

**Performance:** We report the performance of SaNN for trajectory prediction, simplicial closure prediction and graph classification in Tables 1, 2, and 3, respectively. We also provide the per epoch run-time values (in seconds) within brackets. We experimentally observe that SaNN is almost as effective as the existing SNN models, which successively update the features of simplices at every layer, while being several times faster.

**Insights:** For trajectory prediction, SaNN is observed to outperform the projection-based method and has competitive performance as the existing SNN models on all the datasets. The good performance of SaNN also signifies its effective use of the orientations of flows for trajectory prediction. We apply the existing SNN models and SaNN for simplicial closure prediction. The deep models are observed to perform exceptionally better than logistic regression. Of the three deep models, namely, MPSN, SCNN, and SaNN, MPSN is observed to have the best performance on some of the smaller datasets, namely, *high-school,* and *primary-school* datasets. However, on datasets with a large number of simplices, due to the dependence of each layer of the existing SNN models on, approximately, the square of the number of simplices, the existing SNN variants quickly run out of memory. SaNN, on the other hand, performs competitively with the existing SNN models while being many times faster. The computational savings of SaNN are the most evident for this application as the

---

[5]Details about the experimental setup, datasets, attributes, hyperparameters, evaluation metrics, training, validation, and test splits for the three tasks are provided in Appendix H.

Table 1: Trajectory prediction accuracies on various datasets. The first and the second best performances are highlighted in **red** and **blue**, respectively. The values within parentheses are the average per epoch run-time values (in seconds). The first terms in the runtime values of SaNN correspond to the precomputation times the second terms to the per epoch training times. Run-time values of non-deep baseline are indicated by $-$. The best run-time values are highlighted in bold.

| DATASET | OCEAN | SYNTHETIC | PLANAR | MESH |
|---|---|---|---|---|
| PROJECTION | $27.15 \pm 0.0$ 
 $-$ | $52.0 \pm 0.0$ 
 $-$ | $\mathbf{100.0 \pm 0.0}$ 
 $-$ | $30.9 \pm 0.0$ 
 $-$ |
| SCoNe (RODDENBERRY ET AL., 2021) | $\mathbf{30.0 \pm 0.6}$ 
 (0.4) | $\mathbf{55.4 \pm 1.1}$ 
 (3.2) | $97.0 \pm 1.9$ 
 (30.2) | $\mathbf{98.5 \pm 1.9}$ 
 (18.9) |
| SCNN (YANG ET AL., 2022A) | $28.5 \pm 0.6$ 
 (1.9) | $50.5 \pm 1.0$ 
 (4.2) | $\mathbf{97.6 \pm 2.0}$ 
 (36.3) | $98.1 \pm 2.0$ 
 (29.8) |
| SaNN | $\mathbf{35.0 \pm 0.7}$ 
 **(0.01, 0.1)** | $\mathbf{54.9 \pm 1.1}$ 
 **(0.06, 2.5)** | $97.3 \pm 1.9$ 
 **(1.22, 26.8)** | $\mathbf{98.3 \pm 2.0}$ 
 **(0.39, 8.1)** |

Table 2: Relative AUC-PR values for simplicial closure prediction. The first and the second best performances are highlighted in **red** and **blue**, respectively. The values within parentheses are the average per epoch run-time values (in seconds). The first terms in the runtime values of SaNN correspond to the precomputation times and the second terms to the per epoch training times. The best run-time values are highlighted in bold. Out-of-memory results are indicated by −.

| DATASET | ENRON | HIGH-SCHOOL | PRIMARY-SCHOOL | NDC-CLASSES | MATH-SX |
|---|---|---|---|---|---|
| RANDOM BASELINE (RB) | 0.0537 | 0.0112 | 0.0105 | 0.2190 | 0.0202 |
| LOG. REG./RB | $0.55 \pm 0.0$ | $0.59 \pm 0.0$ | $1.79 \pm 0.0$ | $2.32 \pm 0.3$ | $0.65 \pm 0.0$ |
| MPSN (BODNAR ET AL., 2021)/RB | $14.51 \pm 0.1$ (255) | $30.83 \pm 0.0$ (413) | $33.05 \pm 0.0$ (3499) | − | − |
| SCNN (YANG ET AL., 2022A)/RB | $14.17 \pm 0.0$ (17) | $20.53 \pm 0.0$ (401) | $20.19 \pm 0.0$ (1891) | − | − |
| SaNN/RB | $15.45 \pm 0.0$ **(0.01, 3)** | $30.22 \pm 0.0$ **(0.05, 112)** | $32.89 \pm 0.0$ **(0.76, 916)** | $2.79 \pm 0.0$ **(0.26, 13)** | $6.88 \pm 0.0$ **(95.91, 52883)** |

Table 3: Graph classification accuracies on various datasets. The first and the second best performances are highlighted in **red** and **blue**, respectively. The values within parentheses are the average per epoch run-time values (in seconds). The first terms in the runtime values of SPIN and SaNN correspond to the precomputation times and the second terms to the per epoch training times. The best run-time values are highlighted in bold.

| DATASET | PROTEINS | NCI1 | IMDB-B | REDDIT-B | REDDIT-M |
|---|---|---|---|---|---|
| MPSN (BODNAR ET AL., 2021) | $76.5 \pm 3.4$ (33) | $82.8 \pm 2.2$ (292) | $75.6 \pm 3.2$ (46) | $92.2 \pm 1.0$ (242) | $57.3 \pm 1.6$ (1119) |
| DGCNN (PHAN ET AL., 2018) | $72.9 \pm 3.5$ (21) | $76.4 \pm 1.7$ (218) | $69.2 \pm 3.0$ (19) | $87.8 \pm 2.5$ (231) | $49.2 \pm 1.2$ (353) |
| GIN (LESKOVEC & JEGELKA, 2019) | $73.3 \pm 4.5$ (19) | $80.0 \pm 1.4$ (171) | $71.2 \pm 3.9$ (17) | $89.9 \pm 1.9$ (190) | $56.1 \pm 1.7$ (241) |
| GraphSAGE (HAMILTON ET AL., 2017) | $73.0 \pm 4.5$ (17) | $76.0 \pm 1.8$ (167) | $68.8 \pm 4.5$ (15) | $84.3 \pm 1.9$ (188) | $50.0 \pm 1.3$ (219) |
| SPIN (DOSHI & CHEPURI, 2022) | $75.6 \pm 4.5$ **(1.6, 0.3)** | $74.0 \pm 1.7$ **(4, 38)** | $71.1 \pm 5.0$ **(0.3, 7)** | $88.4 \pm 2.5$ **(19, 25)** | $53.8 \pm 1.4$ **(4, 88)** |
| SaNN | $77.6 \pm 2.2$ (3.6, 0.4) | $74.9 \pm 2.2$ (6, 58) | $72.7 \pm 2.1$ (4, 8) | $91.7 \pm 2.7$ (6, 45) | $54.1 \pm 1.6$ (34, 104) |

datasets have several thousands of simplices [cf. Table 6 in Appendix H]. SaNN is observed to have competitive performance as MPSN also for graph classification. Though MPSN has a slightly better performance in terms of the means of the accuracies, in most of the results, the accuracies have significant statistical overlap, i.e., the standard deviations are too high compared to the difference in their means. Hence, comparisons between the best, second best, and others are insignificant. Furthermore, as for the other applications mentioned above, SaNN has the advantage of being almost independent of the number of simplices in the simplicial complexes.

**Additional Insights:** To give additional insights into which aspects of SaNN contribute the most to its competitive performance, we perform ablation studies. We study the effect of features from different depths and observe that the proposed SaNN model that combines features from different depths outperforms ablated SaNN models that use specific neighborhood depths, indicating that although higher-hop-aware features implicitly contain lower-hop information, explicitly combining the information from all the hops has an empirical advantage. This agrees with the theoretical results since the model presented in Section 3, unlike the ablated SaNN models, is provably as expressive as MPSN, suggesting that its embeddings are equally expressive and capable of achieving similar performance in downstream tasks as MPSN. We also observed that local neighborhood information is crucial for all tasks, while higher-hop information seems less relevant. We also studied the effect of features of different orders and observed that combining the transformed features from simplices of different orders results in a much better performance than using only nodes, edges, triangles, or tetrahedrons. Specifically, for simplicial closure prediction, we observe that the constituent edges of open triangles carry the most crucial information about whether a simplex will be formed. This agrees with the observation made in Benson et al. (2018), which states that the tie strengths of the edges in an open triangle positively impact its simplicial closure probability. In graph classification, using only higher-order simplices such as triangles and tetrahedrons leads to poor results, possibly due to the small graph sizes with very few higher-order simplices. These limited higher-order simplices fail to capture the graph structures and distinguish between them. More details from the ablation studies are provided in Appendix J.

## 6 CONCLUSIONS

We have presented a class of simple simplicial neural network models, referred to as simplicial-aware neural networks (SaNN), which are based on the precomputation of simplicial features prior to the training process. We theoretically analyzed the expressive power of SaNN models and provided theoretical conditions under which they discriminate all the simplicial complexes that the SWL test can do. We also provided the conditions under which the class of SaNN models are more powerful than the WL test. We relate the discriminative power of the SaNN models to that of the SWL test by expressing the output of SaNN models as an injective function of the colors assigned by the WL and the SWL tests. We have prescribed viable functions in the SaNN model that result in a simplified yet powerful model that is as expressive as the SWL test. We have demonstrated via experiments that SaNN performs on par with the existing SNN models for trajectory prediction, simplicial closure prediction, and graph classification applications on various benchmark datasets. We thereby show through numerical experiments that SaNN models are computationally inexpensive and well-capture the structure of the simplicial complexes.

## 7 ACKNOLEDGEMENTS

This research was partially supported by a Qualcomm Innovation Fellowship and by the Kotak IISc AI-ML Centre (KIAC).

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

## A   SIMPLICIAL NEURAL NETWORKS

We introduced the geric model for SNNs in Section 2. Here, we present specific details of the aggregation and transformation functions in equation (1) as applied in various SNN models. In MPSN, the first term corresponding to self-embeddings is ignored (i.e., $\mathbf{A}_k^S = \mathbf{0}$), and the next two terms in (1) that aggregate information from the neighboring simplices of the same dimension (here, $k$) are merged as $\mathbf{A}_k \mathbf{H}_k^{(l)} \mathbf{W}_1^{(l)}$, where $\mathbf{A}_k$ is the sum of the upper and the lower Laplacian matrices $\mathbf{B}_k^{\mathrm{T}} \mathbf{B}_k + \mathbf{B}_{k+1} \mathbf{B}_{k+1}^{\mathrm{T}}$ or some normalized version of it. In MPSN, $\mathbf{A}_{k-1}$ and $\mathbf{A}_{k+1}$ are $\mathbf{B}_k^{\mathrm{T}}$ and $\mathbf{B}_k$, respectively. The models SNN (Ebli et al., 2020), SCoNE (Roddenberry et al., 2021), and SCNN (Yang et al., 2022a) aggregate information from neighboring simplices of the same order, considering only the upper and the lower adjacent simplices ($\mathbf{A}_{k-1} = \mathbf{0}$ and $\mathbf{A}_{k+1} = \mathbf{0}$). SNN ignores self-embeddings, while SCoNE and SCNN consider self-embeddings to update the embeddings. In SNN, the aggregation and transformation of aggregated features are performed together for upper and lower adjacent simplices as $\mathbf{A}_k \mathbf{H}_k^{(l)} \mathbf{W}_1^{(l)}$, where $\mathbf{A}_k$ is $\mathbf{B}_k^{\mathrm{T}} \mathbf{B}_k + \mathbf{B}_{k+1} \mathbf{B}_{k+1}^{\mathrm{T}}$. However, in SCoNE and SCNN, the embeddings aggregated from upper and lower adjacent simplices are transformed separately using two different weight matrices $\mathbf{W}_1^{(\mathcal{U},l)}$ and $\mathbf{W}_1^{(\mathcal{L},l)}$, respectively, decoupling the information from the two types of adjacencies. Additionally, SCNN also uses powers of $\mathbf{A}_{k,\mathcal{L}}$ and $\mathbf{A}_{k,\mathcal{L}}$ to aggregate information while SCoNE does not. The model SCoNE, on the other hand, is specialized for the task of trajectory prediction and focuses on simplicial awareness, and permutation and orientation equivariance. The combining function $\psi$ in all the three models MPSN, SNN, and SCoNE is summation. However, in S2CNN (Bunch et al., 2020), the combining function is concatenation. The model proposed in S2CNN focuses on processing data specifically over 2-simplicial complexes and the aggregations in this model are performed using normalized adjacency matrices defined in (Schaub et al., 2020). It should be noted that in all these SNN models, feature aggregation is performed sequentially during training, i.e., the embedding matrix in the $(l + 1)$th layer, denoted by $\mathbf{H}_k^{(l+1)}$, is computed by aggregating the product of the embedding matrices in the $l$th layer, namely, $\mathbf{H}_{k-1}^{(l)}$, $\mathbf{H}_k^{(l)}$, and $\mathbf{H}_{k+1}^{(l)}$, and the associated learnable matrices as in (1). This increases the training time significantly [cf. Fig. 3]. All baseline methods, including MPSN, SCNN, SCoNe, and S2CNN, exhibit simplicial awareness as they utilize the powers of lower-adjacent matrices, upper-adjacent matrices, or Hodge Laplacian matrices, none of which approach zero. However, except for MPSN, these baselines do not match the SWL test in terms of expressivity, as they do not account for feature aggregation from boundary and coboundary simplices, which is considered by the SWL test. Furthermore, since the baselines other than MPSN do not consider aggregation from coboundary simplices, the node embeddings from these models cannot be shown to be more expressive than those from the WL test. We summarize the properties of SaNN and the existing SNN models as follows:

|  | Aggregations | More expressive than WL | As expressive as SWL | Simplicial-awareness |
| --- | --- | --- | --- | --- |
| SaNN | Precomputed | ✓ | ✓ | ✓ |
| MPSN | Sequential | ✓ | ✓ | ✓ |
| ScoNe | Sequential | ✗ | ✗ | ✓ |
| SCNN | Sequential | ✗ | ✗ | ✓ |

## B   COMPARISON OF COMPUTATIONAL COMPLEXITIES OF SNNs AND SaNN

Since we use MLPs to learn from simplicial-aware features, the computational complexity of the proposed SaNN is of the order of computational complexity of MLPs and constant with respect to the number of simplices.

The time complexity of aggregating the neighboring attributes of $k$-simplices during the training of a general SNN model as in (1) with $T$ layers is about $\mathcal{O}\big(T(2N_k^2 D_k + N_k N_{k-1} D_{k-1} + N_k N_{k+1} D_{k+1})\big)$ for a simplicial complex with $N_k$ $k$-simplices with each $k$-simplex having $D_k$-dimensional attributes. The time complexity involved in transforming the aggregated features is about $\mathcal{O}\big(T(3N_k D_k^2 + N_k D_{k-1}^2 + N_k D_{k+1}^2)\big)$. Here, for simplicity, we assume $D_k$-dimensional features in all the hidden layers corresponding to generating embeddings of $k$-simplices. Therefore, the overall time complexity of the existing SNN models is about $\mathcal{O}\big(T((2N_k^2 D_k + N_k N_{k-1} D_{k-1} + N_k N_{k+1} D_{k+1}) + (3N_k D_k^2 + N_k D_{k-1}^2 + N_k D_{k+1}^2))\big)$, which is dominated by the first term containing $2N_k^2 D_k$, $N_k N_{k-1} D_{k-1}$, and $N_k N_{k+1} D_{k+1}$, corresponding to feature aggregation in every layer.

The proposed `SaNN` model performs neighborhood aggregations beforehand, and therefore, its computational complexity during training is only due to the transformation of pre-computed features. Specifically, an `SaNN` model capturing information from $0, \ldots, T$-hop neighborhood of $k$-simplices has a time complexity of $\mathcal{O}(T(3N_k D_k^2 + N_k D_{k-1}^2 + N_k D_{k+1}^2))$, which is negligibly small when compared to the computational complexity of the existing SNN models. This is also reflected in the run-time measurements in Fig. 3.

## C   THE SWL TEST

The simplicial Weisfeiler-Lehman (SWL) test is an iterative algorithm that assigns colors to all the simplices in a simplicial complex by updating the colors of simplices through neighborhood feature aggregation. The details of the algorithm are as follows. Consider two $K$-simplicial comlexes $\mathcal{K}_1$ and $\mathcal{K}_2$. Let the colors assigned by the SWL algorithm in the $t$th iteration to a simplex $\sigma$ be $c_\sigma^{(t)}$. In the $(t+1)$th iteration of the SWL algorithm, each simplex in the two simplicial complexes is assigned a tuple containing the old compressed label of the simplex $c_\sigma^{(t)}$ and the following multisets

$$
\begin{aligned}
c_\mathcal{B}^{(t)}(\sigma) &= \{\!\{ c_\tau^{(t)} \mid \tau \in \mathcal{B}^{(1)}(\sigma) \}\!\}, \\
c_\mathcal{C}^{(t)}(\sigma) &= \{\!\{ c_\tau^{(t)} \mid \tau \in \mathcal{C}^{(1)}(\sigma) \}\!\}, \\
c_\mathcal{L}^{(t)}(\sigma) &= \{\!\{ (c_\tau^{(t)}, c_\delta^{(t)}) \mid \tau \in \mathcal{L}^{(1)}(\sigma) \text{ and } \delta \in \mathcal{B}^{(1)}(\sigma, \tau) \}\!\}, \\
c_\mathcal{U}^{(t)}(\sigma) &= \{\!\{ (c_\tau^{(t)}, c_\delta^{(t)}) \mid \tau \in \mathcal{U}^{(1)}(\sigma) \text{ and } \delta \in \mathcal{C}^{(1)}(\sigma, \tau) \}\!\},
\end{aligned}
\tag{9}
$$

namely, the multisets containing the compressed labels of the boundary adjacent, co-boundary adjacent, lower adjacent, and upper adjacent neighbors of the simplex, respectively, in the $t$th iteration. A new color is then generated by passing the tuple $\bar{c}_\sigma^{(t)} = \left\{ c_\sigma^{(t)}, c_\mathcal{B}^{(t)}(\sigma), c_\mathcal{C}^{(t)}(\sigma), c_\mathcal{L}^{(t)}(\sigma), c_\mathcal{U}^{(t)}(\sigma) \right\}$ through a HASH function as

$$
c_\sigma^{(t+1)} = \text{HASH} \left( \bar{c}_\sigma^{(t)} \right).
\tag{10}
$$

Any two simplices with the same tuple $\bar{c}_\sigma^{(t)}$ will get the same updated color. The algorithm is terminated if the histogram of partitions of simplices partitioned by colors diverges between the two simplicial complexes, and in that case, the simplicial complexes are not isomorphic. It is shown in [4] that `MPSN` is strictly more powerful than the WL test and is as powerful as the SWL test if the neighborhood aggregator functions are injective.

## D   PROOF OF THEOREM 4.1

Throughout the proofs, we use $\mathbf{s}_\sigma^{(t)}$ to represent $\mathbf{S}_k^{(t)}[\sigma]$ and $\mathbf{h}_\sigma^{(t)}$ to represent $\mathbf{H}_k^{(t)}[\sigma]$, whenever the order of $\sigma$ is clear from the context, for notational convenience.

**Lemma D.1.** *For a graph $\mathcal{G} = (\mathcal{V}, \mathcal{E})$, the update of WL coloring of a node $u \in \mathcal{V}$ at iteration $T$ can be represented by the following equation*

$$
c_u^{(T)} = \phi \left( c_u^{(0)}, \left\{\!\!\left\{ c_\tau^{(0)}, \forall \tau \in \{\mathcal{U}^{(1)}(u), \ldots, \mathcal{U}^{(T)}(u)\} \right\}\!\!\right\} \right),
\tag{11}
$$

*where $\phi(.)$ is some injective function and $\mathcal{U}^{(t)}(u)$ denotes the set of upper adjacent $t$-hop neighbors of node $u$.*

*Proof.* We show, by induction, that the update of the WL coloring of a node $u \in \mathcal{V}$ at iteration $T$ is given by (11). The base case, i.e., when $t = 0$, trivially holds as the WL coloring of node $u$ is $c_u^{(0)}$. Let us assume that the update of WL coloring at the $(t-1)$th iteration can be represented by the following equation

$$
c_u^{(t-1)} = \phi_0 \left( c_u^{(0)}, \left\{\!\!\left\{ c_\tau^{(0)}, \forall \tau \in \{\mathcal{U}^{(1)}(u), \ldots, \mathcal{U}^{(t-1)}(u)\} \right\}\!\!\right\} \right),
$$

for some injective function $\phi_0$. By definition, the update of WL coloring (Lehman & Weisfeiler, 1968) of a node $u \in \mathcal{V}$ is given by

$$
c_u^{(t)} = \text{HASH} \left( c_u^{(t-1)}, \left\{\!\!\left\{ c_j^{(t-1)}, \forall j \in \mathcal{U}^{(1)}(u) \right\}\!\!\right\} \right),
\tag{12}
$$

where HASH is an injective function. Using the induction hypothesis, (12) can be written as

$$
\begin{aligned}
c_u^{(t)} = \text{HASH} \Bigg( &\phi_0 \left( c_u^{(0)}, \left\{\!\!\left\{ c_\tau^{(0)}, \forall \tau \in \{\mathcal{U}^{(1)}(u), \ldots, \mathcal{U}^{(t-1)}(u)\} \right\}\!\!\right\} \right), \\
&\left\{\!\!\left\{ \phi_0 \left( c_j^{(0)}, \left\{\!\!\left\{ c_\tau^{(0)}, \forall \tau \in \{\mathcal{U}^{(1)}(j), \ldots, \mathcal{U}^{(t-1)}(j)\} \right\}\!\!\right\} \right), \ \forall j \in \mathcal{U}^{(1)}(u) \right\}\!\!\right\} \Bigg)
\end{aligned}
\tag{13}
$$

As HASH and $\phi_0$ are injective functions, we can express (13) as

$$
\begin{aligned}
c_u^{(t)} &= \phi\left(c_u^{(0)}, \left\{c_\tau^{(0)}, \forall\tau \in \{\mathcal{U}^{(1)}(u),\ldots,\mathcal{U}^{(t-1)}(u)\}\right\}, \left\{c_j^{(0)}, \left\{c_\tau^{(0)}, \forall\tau \in \{\mathcal{U}^{(1)}(j),\ldots,\mathcal{U}^{(t-1)}(j)\}\right\}, \forall j \in \mathcal{U}^{(1)}(u)\right\}\right) \\
&= \phi\left(c_u^{(0)}, \left\{c_\tau^{(0)}, \forall\tau \in \{\mathcal{U}^{(1)}(u),\ldots,\mathcal{U}^{(t-1)}(u)\}\right\}, \left\{c_\tau^{(0)}, \forall\tau \in \{\mathcal{U}^{(1)}(u),\mathcal{U}^{(2)}(u)\ldots,\mathcal{U}^{(t)}(u)\}\right\}\right) \\
&= \phi\left(c_u^{(0)}, \left\{c_\tau^{(0)}, \forall\tau \in \{\mathcal{U}^{(1)}(u),\ldots,\mathcal{U}^{(t)}(u)\}\right\}\right),
\end{aligned}
\tag{14}
$$

for some function $\phi$. Since $\phi$ is a composition of injective multiset functions, it is injective. By induction, the update of WL coloring of a node at the $T$th iteration is given by (11). $\qquad\square$

**Lemma D.2.** *SaNN is atleast as powerful as the WL test in distinguishing non-isomorphic graphs when using a clique-complex lifting if all the functions involved in generating the node embeddings, namely, $f_{0,\mathcal{U}}(\cdot), f_{0,\mathcal{C}}(\cdot), \phi(\cdot), g_0^{(t)}(\cdot),$ and $\Theta_0(\cdot)$ for $t \in 0,\ldots,T$, are injective.*

*Proof.* Let $\mathcal{K}_1$ and $\mathcal{K}_2$ be two $K$-simplicial complexes. Let the embeddings assigned by SaNN to the nodes (0-simplices) $u \in \mathcal{K}_1$ and $v \in \mathcal{K}_2$ be denoted by $\mathbf{h}_u$ and $\mathbf{h}_v$ and the WL coloring at the two nodes be denoted by $c_u$ and $c_v$. We show that for all simplicial complexes $\mathcal{K}_1$ and $\mathcal{K}_2$ and for all the nodes $u \in \mathcal{K}_1$ and $v \in \mathcal{K}_2$, $\mathbf{h}_u = \mathbf{h}_v$ implies $c_u = c_v$. This proves that the WL isomorphism test fails to assign different embeddings to different nodes wherever SaNN fails. In other words, SaNN is atleast as powerful as the WL test in distinguishing non-isomorphic graphs.

Let us denote the $T$-hop neighborhood information capturing node embedding of node $w$ generated using SaNN for a graph with a clique-complex lifting by

$$
\mathbf{h}_w^{(T)} = \Theta_0\left(\mathbf{s}_w^{(0)}, \cdots, \mathbf{s}_w^{(T)}\right),
\tag{15}
$$

where $\mathbf{s}_w^{(i)} = g_0^{(i)}\left(\phi\left(f_{0,\mathcal{U}}\left(\mathbb{X}_{0,\mathcal{U}}^{(i)}[w]\right), f_{0,\mathcal{C}}\left(\mathbb{X}_{1,\mathcal{C}}^{(i)}[w]\right)\right)\right)$ for $i \in \{0,\ldots,t\}$.

We now prove that the WL coloring of two nodes $u \in \mathcal{K}_1$ and $v \in \mathcal{K}_2$ is the same whenever SaNN assigns same embeddings to the nodes. With the assumption that $\Theta_0, g_0^{(t)}, \phi, f_{0,\mathcal{U}}$ and $f_{0,\mathcal{C}}, \forall t \in \{0,\ldots,T\}$, are injective, $\mathbf{h}_w^{(T)}$ can be written as

$$
\begin{aligned}
\mathbf{h}_w^{(T)} &= \Theta_0\left(g_0^{(0)}\left(\phi\left(f_{0,\mathcal{U}}\left(\mathbb{X}_{0,\mathcal{U}}^{(0)}[w]\right)\right)\right), \ldots, g_0^{(T)}\left(\phi\left(f_{0,\mathcal{U}}\left(\mathbb{X}_{0,\mathcal{U}}^{(T)}[w]\right), f_{0,\mathcal{C}}\left(\mathbb{X}_{1,\mathcal{C}}^{(T)}[w]\right)\right)\right)\right) \\
&= \bar\Theta_0\left(c_w^{(0)}, \left\{\!\!\left\{c_\tau^{(0)}, \forall\tau \in \mathcal{U}^{(1)}(w)\right\}\!\!\right\}, \mathbb{X}_{1,\mathcal{C}}^{(1)}[w], \ldots, \left\{\!\!\left\{c_\tau^{(0)}, \forall\tau \in \mathcal{U}^{(T)}(w)\right\}\!\!\right\}, \mathbb{X}_{1,\mathcal{C}}^{(T)}[w]\right),
\end{aligned}
\tag{16}
$$

where $\bar\Theta_0$ is a composition of injective functions and therefore is an injective function itself. Here, the second equality follows because of the assumption that the WL coloring of nodes is initialized with the input features at the nodes as $c_w^{(0)} = \mathbf{X}_0[w]$. If the node embeddings of the nodes $u$ and $v$ are equal, i.e., if $\mathbf{h}_u^{(0)} = \mathbf{h}_v^{(0)}$, the tuple of inputs, i.e., $\left(c_w^{(0)}, \left\{\!\!\left\{c_\tau^{(0)}, \forall\tau \in \mathcal{U}^{(1)}(w)\right\}\!\!\right\}, \mathbb{X}_{1,\mathcal{C}}^{(1)}[w], \ldots, \left\{\!\!\left\{c_\tau^{(0)}, \forall\tau \in \mathcal{U}^{(T)}(w)\right\}\!\!\right\}, \mathbb{X}_{1,\mathcal{C}}^{(T)}[w]\right)$ for $w = u$ and $w = v$, are equal since $\bar\Theta_0$ is an injective function. From Lemma D.1, we notice that the tuples $\left(c_w^{(0)}, \left\{\!\!\left\{c_\tau^{(0)}, \forall\tau \in \{\mathcal{U}^{(1)}(w),\ldots,\mathcal{U}^{(T)}(w)\}\right\}\!\!\right\}\right)$ for $w = u$ and $w = v$, respectively, are the arguments that the WL HASH function takes as input to compute the colors of the nodes $u$ and $v$ at iteration $T$. By the equality of these tuples, we obtain $c_u^{(T)} = c_v^{(T)}$. $\qquad\square$

The above lemma shows that through its node level embeddings, SaNN distinguishes all the simplicial complexes that the WL test can distinguish through its colors. We have, however, ignored the information from higher-order embeddings that SaNN incorporates through co-boundary adjacencies. These embeddings can add more information and help differentiate graphs better. Since it is proved that SaNN is atleast as powerful as the WL test in distinguishing non-isomorphic graphs, it is now sufficient to show an example of a pair of graphs for which the WL test fails to distinguish but the higher-order information in SaNN enables it to differentiate the graphs. Fig. 4 shows an example of such a pair of graphs. The histogram of colors assigned by the WL test and the histogram of embeddings assigned by SaNN at the first three iterations are shown next to the two graphs. The histogram of colors assigned by the WL test to both the graphs are the same. However, SaNN with $T = 3$, i.e., with 2-hop neighborhood aggregation, generates different embeddings to the two graphs. This is because for the graph in Fig. 4(a), SaNN incorporates the information from four triangles in the 2-hop co-boundary of the orange nodes, while in Fig. 4(b), there is only one triangle in the 2-hop neighbood of the orange nodes. e proved that SaNN is atleast as powerful as the WL test and have now shown an example where the WL test fails to distinguish a pair of graphs that SaNN correctly distinguishes as being two different graphs. This proves that SaNN is more powerful than the WL test in distinguishing non-isomorphic graphs.

# E    PROOF OF THEOREM 4.2

To prove Theorem 4.2, we first present a simpler alternative update to the SWL coloring (Bodnar et al., 2021) without sacrificing the expressive power of the SWL test (more details about the test are in Appendix C).

**Lemma E.1.** *For a simplicial complex $\mathcal{K}$, the update of SWL coloring of a simplex $\sigma \in \mathcal{K}$ at iteration t can be represented by the following equation*

$$c_\sigma^{(t)} = \psi\left(c_\sigma^{(t-1)}, \left\{c_j^{(t-1)}, \forall j \in \mathcal{B}^{(1)}(\sigma)\right\}, \left\{c_j^{(t-1)}, \forall j \in \mathcal{C}^{(1)}(\sigma)\right\}, \left\{c_j^{(t-1)}, \forall j \in \mathcal{U}^{(1)}(\sigma)\right\}, \left\{c_j^{(t-1)}, \forall j \in \mathcal{L}^{(1)}(\sigma)\right\}\right).$$
(17)

*for some injective function $\psi$.*

*Proof.* Let us denote the iterative coloring algorithm whose update expression in the $t$th update is (17) by SWL-$\psi$. We prove the equivalence in expressive powers of SWL-$\psi$ and the SWL test (Bodnar et al., 2021) by proving that SWL-$\psi$ is as expressive as MPSN, which is proved to have the same expressive power as the SWL test presented in Bodnar et al. (2021).

The expression for the embeddings of a simplex $\sigma$ obtained using $t$-layers of MPSN is given by

$$\mathbf{z}_\sigma^{(t+1)} = U\left(\mathbf{z}_\sigma^{(t)}, m_\mathcal{B}^{(t)}(\sigma), m_\mathcal{C}^{(t)}(\sigma), m_\mathcal{L}^{(t)}(\sigma), m_\mathcal{U}^{(t)}(\sigma)\right),$$
(18)

where

$$m_\mathcal{B}^{(t)}(\sigma) = \text{AGG}_{\tau \in \mathcal{B}(\sigma)}\left(m_\mathcal{B}\left(\mathbf{z}_\sigma^{(t)}, \mathbf{z}_\tau^{(t)}\right)\right), m_\mathcal{C}^{(t)}(\sigma) = \text{AGG}_{\tau \in \mathcal{C}(\sigma)}\left(m_\mathcal{C}\left(\mathbf{z}_\sigma^{(t)}, \mathbf{z}_\tau^{(t)}\right)\right)$$

$$m_\mathcal{L}^{(t)}(\sigma) = \text{AGG}_{\tau \in \mathcal{L}(\sigma)}\left(m_\mathcal{L}\left(\mathbf{z}_\sigma^{(t)}, \mathbf{z}_\tau^{(t)}, \mathbf{z}_{\sigma \cap \tau}^{(t)}\right)\right), m_\mathcal{U}^{(t)}(\sigma) = \text{AGG}_{\tau \in \mathcal{U}(\sigma)}\left(m_\mathcal{U}\left(\mathbf{z}_\sigma^{(t)}, \mathbf{z}_\tau^{(t)}, \mathbf{z}_{\sigma \cup \tau}^{(t)}\right)\right).$$
(19)

Here, $\mathbf{z}_{\sigma \cup j}$ denotes the embedding of the simplex that is jointly coboundary adjacent to $\sigma$ and $j$ and $\mathbf{z}_{\sigma \cap j}$ denotes that of the simplex that is jointly boundary adjacent to $\sigma$ and $j$. The functions $m_\mathcal{B}, m_\mathcal{C}, m_\mathcal{L}$ and $m_\mathcal{U}$ combine the self-embedding of $\sigma$ with one of their boundary adjacent, coboundary adjacent, lower adjacent and the common boundary adjacent, and upper adjacent and the common coboundary adjacent simplices, respectively. These combined embeddings from all the simplices in the boundary adjacent, coboundary adjacent, lower adjacent, and upper adjacent neighborhood of $\sigma$ are aggregated using the functions $\text{AGG}_{\tau \in \mathcal{B}(\sigma)}, \text{AGG}_{\tau \in \mathcal{C}(\sigma)}, \text{AGG}_{\tau \in \mathcal{L}(\sigma)}$ and $\text{AGG}_{\tau \in \mathcal{U}(\sigma)}$, respectively. The function $U$ outputs the final embedding of $\sigma$ by taking into account the previous color of the simplex, and the aggregated embeddings $m_\mathcal{B}^{(t+1)}(\sigma), m_\mathcal{C}^{(t+1)}(\sigma), m_\mathcal{L}^{(t+1)}(\sigma)$, and $m_\mathcal{U}^{(t+1)}(\sigma)$.

It is proved in Bodnar et al. (2021) that MPSN with injective aggregators and combining functions is as powerful as the SWL test. It is further shown in Bodnar et al. (2021) that the following simplified version of the model, obtained by dropping off a few adjacencies, is equally expressive:

$$\mathbf{z}_\sigma^{(t+1)} = U\left(\mathbf{z}_\sigma^{(t)}, m_\mathcal{B}^{(t)}(\sigma), m_\mathcal{U}^{(t+1)}(\sigma)\right).$$
(20)

Taking all the aggregators and combining functions in (18) and (19) to be injective, we can compactly represent the output of MPSN at the $t$th layer as

$$\mathbf{z}_\sigma^{(t+1)} = \delta\left(\mathbf{z}_\sigma^{(t)}, \left\{\!\!\left\{\{\mathbf{z}_j^{(t)}, \forall j \in \mathcal{A}(\sigma)\}, \forall \mathcal{A}(\sigma)\right\}\!\!\right\}, \{\!\!\{\mathbf{z}_{\sigma \cap j}^{(t)}, \forall j \in \mathcal{L}(\sigma)\}\!\!\}, \{\!\!\{\mathbf{z}_{\sigma \cup j}^{(t)}, \forall j \in \mathcal{U}(\sigma)\}\!\!\}\right)$$
(21)

where $\mathcal{A}(\sigma) \in \{\mathcal{B}(\sigma), \mathcal{C}(\sigma), \mathcal{L}(\sigma), \mathcal{U}(\sigma)\}$, and $\delta$ is an injective function. Here, we used the fact that the composition of injective functions is injective. Similarly, we can express the equally expressive but simplified version of MPSN given in (20) as

$$\mathbf{z}_\sigma^{(t+1)} = \bar{\delta}\left(\mathbf{z}_\sigma^{(t)}, \left\{\!\!\left\{\{\mathbf{z}_j^{(t)}, \forall j \in \mathcal{A}(\sigma)\}, \forall \bar{\mathcal{A}}(\sigma)\right\}\!\!\right\}, \{\!\!\{\mathbf{z}_{\sigma \cup j}^{(t)}, \forall j \in \mathcal{U}(\sigma)\}\!\!\}\right),$$
(22)

where $\bar{\mathcal{A}}(\sigma) \in \{\mathcal{B}(\sigma), \mathcal{U}(\sigma)\}$, and $\bar{\delta}$ is an injective function. Using these representations of the two alternative definitions of the MPSN model, we now prove that SWL-$\psi$ and MPSN have equal expressiveness.

To prove that SWL-$\psi$ and MPSN are equally expressive, we first prove that if MPSN fails to distinguish two simplices, SWL-$\psi$ also fails to distinguish the two simplices. We then prove that if SWL-$\psi$ fails to distinguish two simplices, MPSN also fails to distinguish the two simplices to prove the equivalence in their expressiveness.

Consider two $k$-simplices $\sigma$ and $\tau$ in two simplicial complexes $\mathcal{K}_1$ and $\mathcal{K}_2$. Let us denote the color assigned by SWL-$\psi$ to a $k$-simplex $\sigma$ at $t = T$th iteration by $c_\sigma^{(T)}$ and the embedding obtained using $t = T$-layers of MPSN by $\mathbf{z}_\sigma^{(T)}$. We now prove that if $\mathbf{z}_\sigma^{(T)} = \mathbf{z}_\tau^{(T)}$, then $c_\sigma^{(T)} = c_\tau^{(T)}$ by induction. This trivially holds for the base case, i.e., for $t = 0$, since initial embeddings for both SWL-$\psi$ and MPSN are the initial features on the simplices.

Assume that if $\mathbf{z}_\sigma^{(t)} = \mathbf{z}_\tau^{(t)}$, then $c_\sigma^{(t)} = c_\sigma^{(t)}$. This, from (21), means that the tuple of inputs to MPSN for obtaining the embeddings of $\sigma$ and $\tau$, denoted by

$$\left( \mathbf{z}_\alpha^{(t)}, \left\{\!\!\left\{ \{\!\{ \mathbf{z}_j^{(t)}, \forall j \in \mathcal{A}(\alpha) \}\!\}, \forall \mathcal{A}(\alpha) \right\}\!\!\right\}, \{\!\{\mathbf{z}_{\alpha \cap j}^{(t)}, \forall j \in \mathcal{L}(\alpha)\}\!\}, \{\!\{\mathbf{z}_{\alpha \cup j}^{(t)}, \forall j \in \mathcal{U}(\alpha)\}\!\} \right),$$

is equal for $\alpha = \sigma$ and $\alpha = \tau$ since $\delta$ is an injective function. By the induction hypothesis, we get

$$\left( c_\sigma^{(t)}, \left\{\!\!\left\{ \{\!\{ c_j^{(t)}, \forall j \in \mathcal{A}(\sigma) \}\!\}, \forall \mathcal{A}(\sigma) \right\}\!\!\right\}, \{\!\{c_{\alpha \cap j}^{(t)}, \forall j \in \mathcal{L}(\sigma)\}\!\}, \{\!\{c_{\sigma \cup j}^{(t)}, \forall j \in \mathcal{U}(\sigma)\}\!\} \right)$$

to be equal for $\alpha = \sigma$ and $\alpha = \tau$. Since a subset of these two tuples, namely, $\left( c_\alpha^{(t)}, \left\{\!\!\left\{ \{\!\{ c_j^{(t)}, \forall j \in \mathcal{A}(\alpha) \}\!\}, \forall \mathcal{A}(\alpha) \right\}\!\!\right\} \right)$, for $\alpha \in \{\sigma, \tau\}$, go as input to SWL-$\psi$, we see that SWL-$\psi$ takes the same set of inputs for updating the colors of $\sigma$ and $\tau$ and therefore, outputs the same colors $c_\sigma^{(t+1)}$ and $c_\tau^{(t+1)}$. By induction, if $\mathbf{z}_\sigma^{(T)} = \mathbf{z}_\tau^{(T)}$, then $c_\sigma^{(T)} = c_\tau^{(T)}$.

To complete the proof, we now prove that if SWL-$\psi$ fails to distinguish two simplices, MPSN will also fail to distinguish the two simplices, i.e., if $c_\sigma^{(T)} = c_\tau^{(T)}$, then $\mathbf{z}_\sigma^{(T)} = \mathbf{z}_\tau^{(T)}$. We prove this, again, by induction. The base case, which says that if $c_\sigma^{(0)} = c_\tau^{(0)}$, then $\mathbf{z}_\sigma^{(0)} = \mathbf{z}_\tau^{(0)}$, trivially holds. Assume that if $c_\sigma^{(t)} = c_\tau^{(t)}$, then $\mathbf{z}_\sigma^{(t)} = \mathbf{z}_\tau^{(0)}$ holds. If $c_\sigma^{(t)} = c_\tau^{(t)}$, we have that the tuple of inputs to SWL-$\psi$ to compute $c_\sigma^{(t)}$ and $c_\tau^{(t)}$ are equal [cf. (17)]. By the induction hypothesis, this implies that $\left( \mathbf{z}_\alpha^{(t)}, \left\{\!\!\left\{ \{\!\{ \mathbf{z}_j^{(t)}, \forall j \in \mathcal{A}(\alpha) \}\!\}, \forall \mathcal{A}(\alpha) \right\}\!\!\right\} \right)$ is equal for $\alpha = \sigma$ and $\alpha = \tau$. To prove that $\mathbf{z}_\sigma^{(t+1)} = \mathbf{z}_\tau^{(t+1)}$, we observe from (22) that we should prove that the inputs $\left( \mathbf{z}_\alpha^{(t)}, \left\{\!\!\left\{ \{\!\{ \mathbf{z}_j^{(t)}, \forall j \in \mathcal{A}(\alpha) \}\!\}, \forall \bar{\mathcal{A}}(\alpha) \right\}\!\!\right\}, \{\!\{\mathbf{z}_{\alpha \cup j}^{(t)}, \forall j \in \mathcal{U}(\alpha)\}\!\} \right)$ for $\alpha = \sigma$ and $\alpha = \tau$ are equal. Since we have already proved that $\left( \mathbf{z}_\alpha^{(t)}, \left\{\!\!\left\{ \{\!\{ \mathbf{z}_j^{(t)}, \forall j \in \mathcal{A}(\alpha) \}\!\}, \forall \bar{\mathcal{A}}(\alpha) \right\}\!\!\right\} \right)$ for $\alpha = \sigma$ and $\alpha = \tau$ are equal, the only unknown equivalence is that of the multisets $\{\!\{\mathbf{z}_{\alpha \cup j}^{(t)}, \forall j \in \mathcal{U}(\alpha)\}\!\}$ for $\alpha = \sigma$ and $\alpha = \tau$.

To prove that $\mathbf{z}_\sigma^{(t+1)} = \mathbf{z}_\tau^{(t+1)}$, it is, therefore, sufficient to show that the multisets $\{\!\{\mathbf{z}_{\alpha \cup j}^{(t)}, \forall j \in \mathcal{U}(\alpha)\}\!\}$, for $\alpha = \sigma$ and $\alpha = \tau$ are equal. The colors in these multisets correspond to the coboundary adjacent simplices of $\sigma$ and $\tau$, which are also jointly coboundary adjacent to their upper adjacent simplices. Since we have already proved that the multisets of colors of coboundary adjacent simplices of $\sigma$ and $\tau$ are equal, we need to only show that the multiplicity of the color of each coboundary simplex from the two multisets is the same.

For a $k$-simplex $\alpha$, the color corresponding to each coboundary simplex of $\alpha$ appears exactly $k + 1$ times in $\{\!\{\mathbf{z}_{\alpha \cup j}^{(t)}, \forall j \in \mathcal{U}(\alpha)\}\!\}$. For example, for a 1-simplex (edge), the color of each coboundary 2-simplex (triangle) is included in its update through $\{\!\{\mathbf{z}_{\alpha \cup j}^{(t)}, \forall j \in \mathcal{U}(\alpha)\}\!\}$ every time its corresponding upper adjacent edges are included. Since the number of upper adjacent edges sharing a particular triangle with $\sigma$ is 2, the color of the particular triangle will be included twice. This is true for any coboundary adjacent triangle. Generalizing this to $k$-simplices, we know that the number of times each of the coboundary adjacent simplices of the two $k$-simplices $\sigma$ and $\tau$ are included in $\{\!\{\mathbf{z}_{\alpha \cup j}^{(t)}, \forall j \in \mathcal{U}(\alpha)\}\!\}$ is $k + 1$ for $\alpha = \sigma$ and $\alpha = \tau$. This proves the equivalence of the two multisets $\{\!\{\mathbf{z}_{\alpha \cup j}^{(t)}, \forall j \in \mathcal{U}(\alpha)\}\!\}$ for $\alpha = \sigma$ and $\alpha = \tau$. Therefore, the inputs to MPSN are the same [cf. (22)], and hence the outputs, denoted by $\mathbf{z}_\sigma^{(t+1)}$ and $\mathbf{z}_\tau^{(t+1)}$, will be the same. By induction $c_\sigma^{(T)} = c_\tau^{(T)}$ implies $\mathbf{z}_\sigma^{(T)} = \mathbf{z}_\tau^{(T)}$.

We proved that if $\mathbf{z}_\sigma^{(T)} = \mathbf{z}_\tau^{(T)}$, then $c_\sigma^{(T)} = c_\tau^{(T)}$, and if $c_\sigma^{(T)} = c_\tau^{(T)}$, then $\mathbf{z}_\sigma^{(T)} = \mathbf{z}_\tau^{(T)}$. This proves that SWL-$\psi$ is as expressive as MPSN and hence, will serve as a simpler alternative to the SWL test. □

**Lemma E.2.** *For a simplicial complex $\mathcal{K}$, the update of SWL coloring of a simplex $\sigma \in \mathcal{K}$ at iteration $T$ can be represented by the following equation*

$$c_\sigma^{(T)} = \phi \left( c_\sigma^{(0)}, \left\{\!\!\left\{ \left\{\!\!\left\{ \{\!\{ c_\tau^{(0)}, \forall \tau \in \mathcal{A}^{(i)}(\sigma) \}\!\}, \forall \mathcal{A}^{(i)} \right\}\!\!\right\}, \forall i = 1, \dots, T \right\}\!\!\right\} \right), \tag{23}$$

*where $\mathcal{A}^{(i)}(\sigma) \in \{\mathcal{B}^{(i)}(\sigma), \mathcal{C}^{(i)}(\sigma), \mathcal{L}^{(i)}(\sigma), \mathcal{U}^{(i)}(\sigma)\}$, for some injective function $\phi$.*

*Proof.* We show that the update of the SWL-$\psi$ coloring, equivalently, the SWL coloring, of a simplex $\sigma \in \mathcal{K}$ at iteration $T$ is given by the equation in Lemma E.2 by induction. The base case, i.e., when $t = 0$, trivially holds as the SWL coloring of simplex $\sigma$ is $c_\sigma^{(0)}$. Let us assume that the update of SWL coloring at the $(t-1)$th iteration can be represented by the following equation

$$c_\sigma^{(t-1)} = \phi \left( c_\sigma^{(0)}, \left\{\!\!\left\{ \left\{\!\!\left\{ \{\!\{ c_\tau^{(0)}, \forall \tau \in \mathcal{A}^{(i)}(\sigma) \}\!\}, \forall \mathcal{A}^{(i)} \right\}\!\!\right\}, \forall i = 1, \dots, t-1 \right\}\!\!\right\} \right), \tag{24}$$

for some injective function $\phi$. Using the induction hypothesis, (17) can be expanded by substituting (24) in (17). Substituting (24) in the second term of (17), results in

$$\left\{\!\!\left\{ c_j^{(t-1)}, \forall j \in \mathcal{B}^{(1)}(\sigma) \right\}\!\!\right\} = \left\{\!\!\left\{ \phi\left(c_j^{(0)}, \left\{\!\!\left\{ \left\{\!\!\left\{ c_\tau^{(0)}, \forall \tau \in \mathcal{A}^{(i)}(\sigma) \right\}\!\!\right\}, \forall \mathcal{A}^{(i)}(\sigma) \right\}\!\!\right\}, \forall i = 1, \ldots, t-1 \right\}\!\!\right\}\right), \forall j \in \mathcal{B}^{(1)}(\sigma) \right\}\!\!\right\}.$$

It can be simplified further by using some injective function $\bar{\phi}$ as

$$\left\{\!\!\left\{ c_j^{(t-1)}, \forall j \in \mathcal{B}^{(1)}(\sigma) \right\}\!\!\right\} = \bar{\phi}\left( \left\{\!\!\left\{ c_\tau^{(0)} \, \forall \tau \in \mathcal{B}^{(1)}(\sigma) \right\}\!\!\right\}, \left\{\!\!\left\{ c_\tau^{(0)}, \forall \tau \in \left\{ \mathcal{B}(\mathcal{A}^{(i)}(\sigma)) \right\} \right\}\!\!\right\}, \forall \mathcal{A}^{(i)}, \forall i = 1, \ldots, t-1 \right\}\!\!\right\}\right).$$

By substituting for $\left\{ \mathcal{B}(\mathcal{A}^{(i)}(\sigma)) \right\}, \forall \mathcal{A}^{(i)}$ with $\mathcal{B}^{(i+1)}(\sigma)$, we obtain

$$\left\{\!\!\left\{ c_j^{(t-1)}, \forall j \in \mathcal{B}^{(1)}(\sigma) \right\}\!\!\right\} = \bar{\phi}\left( \left\{\!\!\left\{ c_\tau^{(0)} \, \forall \tau \in \mathcal{B}^{(1)}(\sigma) \right\}\!\!\right\}, \left\{\!\!\left\{ c_\tau^{(0)}, \forall \tau \in \mathcal{B}^{(i+1)} \right\}\!\!\right\}, \forall i = 1, \ldots, t-1 \right\}\!\!\right\}\right).$$

We repreat the procedure for the other terms in (17) to obtain the following expression for $c_\sigma^{(t)}$

$$c_\sigma^{(t)} = \mathrm{HASH}\left( \phi\left(c_\sigma^{(0)}, \left\{\!\!\left\{ \left\{\!\!\left\{ c_\tau^{(0)}, \forall \tau \in \mathcal{A}^{(i)}(\sigma) \right\}\!\!\right\}, \forall \mathcal{A}^{(i)}(\sigma) \right\}\!\!\right\}, \forall i = 1, \ldots, t-1 \right\}\!\!\right\}\right),$$
$$\left\{\!\!\left\{ \bar{\phi}\left( \left\{\!\!\left\{ c_\tau^{(0)} \, \forall \tau \in \mathcal{A}^{(1)}(\sigma) \right\}\!\!\right\}, \left\{\!\!\left\{ c_\tau^{(0)}, \forall \tau \in \mathcal{A}^{(i+1)} \right\}\!\!\right\}\right), \forall \mathcal{A}^{(i)} \right\}\!\!\right\}, \forall i = 1, \ldots, t-1 \right\}\!\!\right\}\right). \quad (25)$$

Since $\phi$ and $\bar{\phi}$ are injective, we futher simplify the expression for $c_\sigma^{(t)}$ as

$$c_\sigma^{(t)} = \Delta\left( c_\sigma^{(0)}, \left\{\!\!\left\{ \left\{\!\!\left\{ c_\tau^{(0)}, \forall \tau \in \mathcal{A}^{(i)}(\sigma) \right\}\!\!\right\}, \forall \mathcal{A}^{(i)} \right\}\!\!\right\}, \forall i = 1, \ldots, t \right\}\!\!\right\}\right). \quad (26)$$

for some function $\Delta(\cdot)$, which is a composition of injective functions and hence is injective. By induction, the update of the SWL coloring of a simplex at the $T$th iteration is given by the equation in Lemma E.2. $\square$

**Lemma E.3.** *SaNN is at most as powerful as the SWL test in distinguishing non-isomorphic simplicial complexes.*

*Proof.* Let $\mathcal{K}_1$ and $\mathcal{K}_2$ be two $K$-simplicial complexes. Let the embeddings assigned by SaNN to the simplices $\sigma \in \mathcal{K}_1$ and $\tau \in \mathcal{K}_2$ be denoted by $\mathbf{h}_\sigma$ and $\mathbf{h}_\tau$ and the SWL coloring at the two simplices be denoted by $c_\sigma$ and $c_\tau$. We show that for all the simplices $\sigma \in \mathcal{K}_1$ and $\tau \in \mathcal{K}_2$, $c_\sigma = c_\tau$ implies $\mathbf{h}_\sigma = \mathbf{h}_\tau$. With this, we prove that SaNN fails to assign different embeddings to different nodes wherever SWL isomorphism test fails. In other words, SaNN is at most as powerful as the SWL test in distinguishing non-isomorphic simplicial complexes.

From Lemma E.2, the update of SWL coloring of a $k$-simplex $\sigma$ in an $K$-simplicial complex $\mathcal{K}$, at iteration $T$ can be represented by the following equation

$$c_\sigma^{(T)} = \phi\left( c_\sigma^{(0)}, \left\{\!\!\left\{ \left\{\!\!\left\{ c_\tau^{(0)}, \forall \tau \in \mathcal{A}^{(i)}(\sigma) \right\}\!\!\right\}, \forall \mathcal{A}^{(i)} \right\}\!\!\right\}, \forall i = 1, \ldots, T \right\}\!\!\right\}\right), \quad (27)$$

where $\phi(\cdot)$ is some injective function.

The embedding of a $k$-simplex $\sigma \in \mathcal{K}$ generated using SaNN capturing information from $T$-hop neighborhood, denoted by $\mathbf{h}_\sigma^{(T)}$, is given by

$$\mathbf{h}_\sigma^{(T)} = \Theta_k\left( \mathbf{s}_\sigma^{(0)}, \cdots, \mathbf{s}_\sigma^{(T)} \right), \quad (28)$$

where

$$\mathbf{s}_\sigma^{(i)} = g_k^{(i)}\left( \phi\left( f_{k,\mathcal{U}}\left( \mathbb{X}_{k,\mathcal{U}}^{(i)}[\sigma] \right), f_{k,\mathcal{L}}\left( \mathbb{X}_{k,\mathcal{L}}^{(i)}[\sigma] \right), f_{k,\mathcal{B}}\left( \mathbb{X}_{k-1,\mathcal{B}}^{(i)}[\sigma] \right), f_{k,\mathcal{C}}\left( \mathbb{X}_{k+1,\mathcal{C}}^{(i)}[\sigma] \right) \right) \right),$$

for $i \in \{0, \ldots, T\}$.

We now prove that by capturing information from $T$-hop neighborhood for simplices of each order, SaNN assigns the same embeddings to two $k$-simplices $\sigma_1 \in \mathcal{K}_1$ and $\sigma_2 \in \mathcal{K}_2$ if the SWL coloring of the simplices is the same at the $T$th iteration. Let us assume that the SWL coloring of two simplices, $\sigma_1$ and $\sigma_2$, after $T$ iterations are equal, i.e., $c_{\sigma_1}^{(T)} = c_{\sigma_2}^{(T)}$. This, from Lemma E.2, implies that the tuple of inputs,

$$\left( c_\sigma^{(0)}, \left\{\!\!\left\{ \left\{\!\!\left\{ c_\tau^{(0)}, \forall \tau \in \mathcal{A}^{(i)}(\sigma) \right\}\!\!\right\}, \forall \mathcal{A}^{(i)} \right\}\!\!\right\}, \forall i = 1, \ldots, T \right\}\!\!\right\}\right)$$

for $\sigma = \sigma_1$ and $\sigma = \sigma_2$ are equal. However, these are the input arguments to generate the SaNN embeddings of simplices $\sigma_1$ and $\sigma_2$ using (28). Therefore, we obtain $\mathbf{h}_{\sigma_1}^{(T)} = \mathbf{h}_{\sigma_2}^{(T)}$. $\square$

As we have already shown that `SaNN` is at most as powerful as the SWL test in Lemma E.3, it is now sufficient to show that `SaNN` is at least as powerful as the SWL test under the condition that the functions involved in generating the embeddings are injective, to show that it is as powerful as SWL. We therefore show that if the functions $f_{k,\mathcal{U}}(\cdot), f_{k,\mathcal{L}}(\cdot), f_{k,\mathcal{B}}(\cdot), f_{k,\mathcal{C}}(\cdot), \phi(\cdot), g_k^{(t)}(\cdot)$, and $\Theta_k(\cdot)$ for $t \in 0, \ldots, T$ and $k \in 0, \ldots, K$ are injective, $\mathbf{h}_{\sigma_1}^{(T)} = \mathbf{h}_{\sigma_2}^{(T)}$ implies $c_{\sigma_1}^{(T)} = c_{\sigma_2}^{(T)}$.

When the functions involved in generating the embeddings are injective, we can simplify the embeddings of a simplex $\sigma$ of order $k$ as follows

$$
\begin{aligned}
\mathbf{h}_\sigma^{(T)} &= \Theta_k \left( g_k^{(t)} \left( \phi \left( f_{k,\mathcal{U}} \left( \mathbb{X}_{k,\mathcal{U}}^{(t)}[\sigma] \right), f_{k,\mathcal{L}} \left( \mathbb{X}_{k,\mathcal{L}}^{(t)}[\sigma] \right), f_{k,\mathcal{B}} \left( \mathbb{X}_{k-1,\mathcal{B}}^{(t)}[\sigma] \right), f_{k,\mathcal{C}} \left( \mathbb{X}_{k+1,\mathcal{C}}^{(t)}[\sigma] \right) \right) \right), \forall t \in 0, \ldots, T \right) \\
&= \Delta \left( \left\{\!\!\left\{ \mathbb{X}_{k,\mathcal{U}}^{(t)}[\sigma], \mathbb{X}_{k,\mathcal{L}}^{(t)}[\sigma], \mathbb{X}_{k-1,\mathcal{B}}^{(t)}[\sigma], \mathbb{X}_{k+1,\mathcal{C}}^{(t)}[\sigma] \right\}\!\!\right\}, \forall t \in 0, \ldots, T \right) \\
&= \Delta \left( \left\{\!\!\left\{ \left\{\!\!\left\{ c_\tau, \forall \tau \in \mathcal{A}^{(t)}(\sigma) \right\}\!\!\right\}, \forall \mathcal{A}^{(t)} \right\}\!\!\right\}, \forall t \in 0, \ldots, T \right) \quad (29)
\end{aligned}
$$

for some injective function $\Delta(\cdot)$.

Let us now assume $\mathbf{h}_{\sigma_1}^{(T)} = \mathbf{h}_{\sigma_2}^{(T)}$. This implies that the tuples of inputs

$$
\left( \left\{\!\!\left\{ \left\{\!\!\left\{ c_\tau, \forall \tau \in \mathcal{A}^{(t)}(\sigma) \right\}\!\!\right\}, \forall \mathcal{A}^{(t)} \right\}\!\!\right\}, \forall t \in 0, \ldots, T \right)
$$

for $\sigma = \sigma_1$ and $\sigma = \sigma_2$ are equal. Since these are the input arguments to compute the SWL coloring updates of the simplices $\sigma_1$ and $\sigma_2$ at the $T$th iteration, we obtain $c_{\sigma_1}^{(T)} = c_{\sigma_2}^{(T)}$.

## F    REMARK ON INJECTIVITY OF DEGREE-BASED WEIGHTED SUMMATION

Here, we give a generalized case where the degree-based weighted sum aggregator is not injective. That is, we provide a case where the proposed architecture assigns the same partial updates to $k$-simplices in simplicial complexes with different structures.

**Remark 1.** *Suppose all the $k$-simplices in a simplicial complex have the same number of upper (respectively, lower, boundary, and co-boundary) adjacent neighbors, then we refer to this structure as upper (respectively, lower, boundary, and co-boundary) $k$-regular simplicial complex. Consider two upper (respectively, lower, boundary, and co-boundary) $k$-regular $K$-simplicial complexes $\mathcal{K}_1$ and $\mathcal{K}_2$. Let $n_{\mathcal{K}_i,\mathcal{U}}^{(k)}$ (respectively, $n_{\mathcal{K}_i,\mathcal{L}}^{(k)}, n_{\mathcal{K}_i,\mathcal{B}}^{(k)}$, and $n_{\mathcal{K}_i,\mathcal{C}}^{(k)}$), for $i = 1, 2$ denote the number of upper (respectively, lower, boundary, and co-boundary) adjacent simplices of all the $k$-simplices in $\mathcal{K}_i$. Assume that for the two simplicial complexes, the input features of all the $k$-simplices are the same. When $n_{\mathcal{K}_1,\mathcal{U}}^{(k)} \neq n_{\mathcal{K}_2,\mathcal{U}}^{(k)}$ (respectively, $n_{\mathcal{K}_1,\mathcal{L}}^{(k)} \neq n_{\mathcal{K}_2,\mathcal{L}}^{(k)}$, $n_{\mathcal{K}_1,\mathcal{B}}^{(k)} \neq n_{\mathcal{K}_2,\mathcal{B}}^{(k)}$, and $n_{\mathcal{K}_1,\mathcal{C}}^{(k)} \neq n_{\mathcal{K}_2,\mathcal{C}}^{(k)}$), the structures of the two simplicial complexes are different. However, the partial embeddings of $k$-simplices obtained by upper (respectively, lower, boundary, and co-boundary) neighborhood aggregation using normalized incidence matrices defined above for the two non-isomorphic simplicial complexes will be the same.*

Let us consider an illustrative example to better understand the above remark. The two simplicial complexes in Fig. 5 are lower 2-regular simplicial complexes with $n_{\mathcal{K}_1,\mathcal{L}}^{(2)} = 3$ and $n_{\mathcal{K}_2,\mathcal{L}}^{(2)} = 2$. The simplicial complex in Fig. 5 (a) has a tetrahedron, each face of which is a triangle with three lower adjacent triangles. All the other triangles in Fig. 5 (a) also have three lower adjacent triangles. All the triangles in Fig. 5 (b) have two lower adjacent triangles. Assume that the features over the 2-simplices are all $a$. Despite having different structures, the partial embeddings of all the 2-simplices aggregated from their lower adjacent neighbors in the two non-isomorphic simplicial complexes are $a$ if degree-based weighted summation aggregator is used.

For the case in Remark 1, given the injectivity of $\phi$, the proposed architecture assigns the same histogram of $k$-simplicial embeddings to the two simplicial complexes if the two simplicial complexes are simultaneously upper regular, lower regular, boundary regular, and co-boundary regular. Even if the two simplicial complexes do not have the regular structure along one kind of neighborhood, i.e., upper, lower, boundary, or co-boundary, the proposed architecture will be able to distinguish the two structures based on that kind of neighborhood. For example, consider the lower 2-regular simplicial complexes in Fig. 5. Any of the triangles, say $\tau_1$, in the tetrahedron will be assigned the same partial embedding as any of the other triangles in Fig. 5 (a), say $\sigma_1$, and as any of the triangles in Fig. 5 (b), say $\sigma_2$, after aggregation from lower adjacent simplices. However, the partial embedding computed by aggregating information from the upper adjacent triangles helps `SaNN` assign a different partial embedding to the triangles in the tetrahedron, thereby assigning different final embeddings to $\tau_1$ and $\sigma_1$ (or $\sigma_2$). In spite of assigning the same lower adjacent neighborhood aggregated partial embeddings, `SaNN` will, therefore, be able to distinguish the two simplicial complexes with the degree-based weighted summation aggregator using the histogram of embeddings of only the 2-simplices. The cases where the simplicial complexes are regular along all kinds of neighborhoods for simplices of all orders are very rare and degree-based weighted summation is, therefore, almost always injective.

## G   PROOFS OF PROPERTIES OF SANN

Here, we prove the properties of SaNN presented in Section 4.2.

### G.1   PERMUTATION EQUIVARIANCE

Consider a $K$-simplical complex $\mathcal{K}$. Let us define a set of permutation matrices $\mathcal{P} = \{\mathbf{P}_0, \ldots, \mathbf{P}_K\}$ acting on the input feature matrices $\mathbf{X} = \{\mathbf{X}_k \in \mathbb{R}^{N_k \times D_k}\}_{k=0}^{K}$ and the boundary matrices $\mathbf{B} = \{\mathbf{B}_k \in \mathbb{R}^{N_{k-1} \times N_k}\}_{k=1}^{K}$. The permuted input features are represented by $\mathcal{P}\mathbf{X} = (\mathbf{P}_0\mathbf{X}_0, \ldots, \mathbf{P}_N\mathbf{X}_K)$ and the permuted boundary matrices are represented by $\mathcal{P}\mathbf{B}\mathcal{P}^{\mathrm{T}} = (\mathbf{P}_0\mathbf{B}_1\mathbf{P}_1^{\mathrm{T}}, \ldots, \mathbf{P}_{K-1}\mathbf{B}\mathbf{P}_K^{\mathrm{T}})$. We now prove that if the input features to SaNN are permuted using $\mathcal{P}$, SaNN produces an output that is permuted by the same permutation $\mathcal{P}$. Mathematically, if the embedding matrices of $k$-simplices combined from different hops of neighborhood output by SaNN are denoted by $\mathrm{SaNN}(\mathbf{X}, \mathbf{B}) = \{\mathbf{H}_k \in \mathbb{R}^{N_k \times \bar{D}_k}\}_{k=0}^{K}$, we show that $\mathrm{SaNN}(\mathcal{P}\mathbf{X}, \mathcal{P}\mathbf{B}\mathcal{P}^{\mathrm{T}}) = \mathcal{P}\mathrm{SaNN}(\mathbf{X}, \mathbf{B}) = \mathcal{P}\mathbf{H} = (\mathbf{P}_0\mathbf{H}_0, \ldots, \mathbf{P}_K\mathbf{H}_K)$.

*Proof.* The aggregated feature matrix of a $k$-dimensional simplex output by the $t$th branch of SaNN corresponding to $t$-hop neighborhood aggregation is given by

$$\mathbf{X}_k^{(t)} = \left[\mathbf{A}_{k,\mathcal{U}}\mathbf{X}_k^{(t-1)}, \mathbf{A}_{k,\mathcal{L}}\mathbf{X}_k^{(t-1)}, \mathbf{A}_{k-1,\mathcal{B}}\mathbf{X}_{k-1}^{(t-1)}, \mathbf{A}_{k+1,\mathcal{C}}\mathbf{X}_{k+1}^{(t-1)}\right]^{\mathrm{T}}, \tag{30}$$

where $t \in \{1, \ldots, T\}$, $k \in \{0, \ldots, K\}$, and the operators are as defined in Section V of the manuscript. We first prove that if the input features to SaNN are permuted using $\mathcal{P}$, then the 1-hop aggregated features are permuted by the same $\mathcal{P}$. We then prove, by induction, that if the input features to SaNN are permuted using $\mathcal{P}$, then the aggregated features of any depth are permuted by the same $\mathcal{P}$.

For the base case, when $t = 1$, the aggregated feature matrix output by 1-hop neighbourhood aggregation when the input is $(\mathcal{P}\mathbf{X}, \mathcal{P}\mathbf{B}\mathcal{P}^{\mathrm{T}})$ is the following

$$\bar{\mathbf{X}}_k^{(1)} = \left[(\mathbf{P}_k\mathbf{B}_{k+1}\mathbf{P}_{k+1}^{\mathrm{T}}\mathbf{P}_{k+1}\mathbf{B}_{k+1}^{\mathrm{T}}\mathbf{P}_k^{\mathrm{T}}) \cdot (\mathbf{P}_k\mathbf{X}_k), (\mathbf{P}_k\mathbf{B}_k^{\mathrm{T}}\mathbf{P}_{k-1}^{\mathrm{T}}\mathbf{P}_{k-1}\mathbf{B}_k\mathbf{P}_k^{\mathrm{T}}) \cdot (\mathbf{P}_k\mathbf{X}_k), \right.$$

$$\left. (\mathbf{P}_k\mathbf{B}_k^{\mathrm{T}}\mathbf{P}_{k-1}^{\mathrm{T}}) \cdot (\mathbf{P}_{k-1}\mathbf{X}_{k-1}), (\mathbf{P}_k\mathbf{B}_{k+1}\mathbf{P}_{k+1}^{\mathrm{T}}) \cdot (\mathbf{P}_{k+1}\mathbf{X}_{k+1})\right]^{\mathrm{T}},$$

$$= \left[\mathbf{P}_k\mathbf{B}_{k+1}\mathbf{B}_{k+1}^{\mathrm{T}}\mathbf{X}_k, \mathbf{P}_k\mathbf{B}_k^{\mathrm{T}}\mathbf{B}_k\mathbf{X}_k, \ \mathbf{P}_k\mathbf{B}_k^{\mathrm{T}}\mathbf{X}_{k-1}, \mathbf{P}_k\mathbf{B}_{k+1}\mathbf{X}_{k+1}\right]^{\mathrm{T}},$$

$$= \mathbf{P}_k\left[\mathbf{B}_{k+1}\mathbf{B}_{k+1}^{\mathrm{T}}\mathbf{X}_k, \mathbf{B}_k^{\mathrm{T}}\mathbf{B}_k\mathbf{X}_k, \ \mathbf{B}_k^{\mathrm{T}}\mathbf{X}_{k-1}, \mathbf{B}_{k+1}\mathbf{X}_{k+1}\right]^{\mathrm{T}},$$

$$= \mathbf{P}_k\mathbf{X}_k^{(1)} \tag{31}$$

for $k = 0, \ldots, K$. Second equality is achieved using the fact that $\mathbf{P}_k\mathbf{P}_k^{\mathrm{T}} = \mathbf{I}, \forall k \in 0, \ldots, K$. This shows that when the inputs $\mathbf{X}$ and $\mathbf{B}$ are permuted by the set of matrices $\mathcal{P}$, the 1-hop aggregated features $\{\mathbf{X}_k^{(1)} \in \mathbb{R}^{N_k \times D_k}\}_{k=0}^{K}$ are also permuted by the same set of permutation matrices.

Let us assume that when the inputs $\mathbf{X}$ and $\mathbf{B}$ are permuted by the set of matrices $\mathcal{P}$, the $(t-1)$-hop aggregated features $\{\mathbf{X}_k^{(t-1)} \in \mathbb{R}^{N_k \times D_k}\}_{k=0}^{K}$ are also permuted by the same set of permutation matrices, i.e., $\bar{\mathbf{X}}_k^{(t-1)} = \mathbf{P}_k\mathbf{X}_k^{(t-1)}$. Using the induction hypothesis, the aggregated feature matrix output by $t$-hop neighbourhood aggregation when the input is $(\mathcal{P}\mathbf{X}, \mathcal{P}\mathbf{B}\mathcal{P}^{\mathrm{T}})$ is the following

$$\bar{\mathbf{X}}_k^{(t)} = \left[(\mathbf{P}_k\mathbf{B}_{k+1}\mathbf{P}_{k+1}^{\mathrm{T}}\mathbf{P}_{k+1}\mathbf{B}_{k+1}^{\mathrm{T}}\mathbf{P}_k^{\mathrm{T}}) \cdot (\bar{\mathbf{X}}_k^{(t-1)}), (\mathbf{P}_k\mathbf{B}_k^{\mathrm{T}}\mathbf{P}_{k-1}^{\mathrm{T}}\mathbf{P}_{k-1}\mathbf{B}_k\mathbf{P}_k^{\mathrm{T}}) \cdot (\bar{\mathbf{X}}_k^{(t-1)}), \right.$$

$$\left. (\mathbf{P}_k\mathbf{B}_k^{\mathrm{T}}\mathbf{P}_{k-1}^{\mathrm{T}}) \cdot (\bar{\mathbf{X}}_{k-1}^{(t-1)}), (\mathbf{P}_k\mathbf{B}_{k+1}\mathbf{P}_{k+1}^{\mathrm{T}}) \cdot (\bar{\mathbf{X}}_{k+1}^{(t-1)})\right]^{\mathrm{T}},$$

$$= \left[(\mathbf{P}_k\mathbf{B}_{k+1}\mathbf{P}_{k+1}^{\mathrm{T}}\mathbf{P}_{k+1}\mathbf{B}_{k+1}^{\mathrm{T}}\mathbf{P}_k^{\mathrm{T}}) \cdot (\mathbf{P}_k\mathbf{X}_k^{(t-1)}), (\mathbf{P}_k\mathbf{B}_k^{\mathrm{T}}\mathbf{P}_{k-1}^{\mathrm{T}}\mathbf{P}_{k-1}\mathbf{B}_k\mathbf{P}_k^{\mathrm{T}}) \cdot (\mathbf{P}_k\mathbf{X}_k^{(t-1)}), \right.$$

$$\left. (\mathbf{P}_k\mathbf{B}_k^{\mathrm{T}}\mathbf{P}_{k-1}^{\mathrm{T}}) \cdot (\mathbf{P}_{k-1}\mathbf{X}_{k-1}^{(t-1)}), (\mathbf{P}_k\mathbf{B}_{k+1}\mathbf{P}_{k+1}^{\mathrm{T}}) \cdot (\mathbf{P}_{k+1}\mathbf{X}_{k+1}^{(t-1)})\right]^{\mathrm{T}},$$

$$= \left[\mathbf{P}_k\mathbf{B}_{k+1}\mathbf{B}_{k+1}^{\mathrm{T}}\mathbf{X}_k^{(t-1)}, \mathbf{P}_k\mathbf{B}_k^{\mathrm{T}}\mathbf{B}_k\mathbf{X}_k^{(t-1)}, \ \mathbf{P}_k\mathbf{B}_k^{\mathrm{T}}\mathbf{X}_{k-1}^{(t-1)}, \mathbf{P}_k\mathbf{B}_{k+1}\mathbf{X}_{k+1}^{(t-1)}\right]^{\mathrm{T}},$$

$$= \mathbf{P}_k\left[\mathbf{B}_{k+1}\mathbf{B}_{k+1}^{\mathrm{T}}\mathbf{X}_k^{(t-1)}, \mathbf{B}_k^{\mathrm{T}}\mathbf{B}_k\mathbf{X}_k^{(t-1)}, \ \mathbf{B}_k^{\mathrm{T}}\mathbf{X}_{k-1}^{(t-1)}, \mathbf{B}_{k+1}\mathbf{X}_{k+1}^{(t-1)}\right]^{\mathrm{T}},$$

$$= \mathbf{P}_k\mathbf{X}_k^{(t)}, \tag{32}$$

where the second equality follows from the induction hypothesis. This proves that the neighborhood aggregation in SaNN is permutation equivariant.

Recall that we choose the transformation functions $g_k^{(t)}(\cdot)$ to be MLPs and the functions that combine embeddings from different hops, i.e., $\Theta_k(\cdot)$, to be concatenation. Both of these operations preserve permutation equivariance and since SaNN is a composition of the permutation equivariant neighborhood aggregation, MLPs and concatenation, it is permutation equivariant. □

## G.2 ORIENTATION EQUIVARIANCE

Consider a $K$-simplical complex $\mathcal{K}$. The change in orientation of a simplex refers to an orientation reversal of a simplex with respect to the neighboring simplices. This can be represented by multiplying the feature vector on the simplex, and the rows and columns corresponding to the simplex in the boundary matrices with -1. To define the change in orientations, as in the case of permutation equivariance, let us define a set of matrices $\mathcal{O} = (\mathbf{O}_0, \ldots, \mathbf{O}_K)$ acting on the input feature matrices and the boundary matrices. The matrices $\mathbf{O}_k, \forall k = 0, \ldots, K$ are diagonal matrices with $(i,i)$th entry equal to $-1$ if the orientation of the $i$th simplex of dimension $k$ is reversed or is equal to 1 otherwise. The input features with a change in orientation are represented by $\mathcal{O}\mathbf{X} = (\mathbf{O}_0\mathbf{X}_0, \ldots, \mathbf{O}_K\mathbf{X}_K)$ and the boundary matrices modified due to the orientation changes are represented by $\mathcal{O}\mathbf{B}\mathcal{O}^{\mathrm{T}} = (\mathbf{O}_0\mathbf{B}_1\mathbf{O}_1^{\mathrm{T}}, \ldots, \mathbf{O}_{K-1}\mathbf{B}\mathbf{O}_K^{\mathrm{T}})$. Mathematically, we show that $\mathrm{SaNN}(\mathcal{O}\mathbf{X}, \mathcal{O}\mathbf{B}\mathcal{O}^{\mathrm{T}}) = \mathcal{O}\mathrm{SaNN}(\mathbf{X}, \mathbf{B}) = \mathcal{O}\mathbf{H} = (\mathbf{O}_0\mathbf{H}_0, \ldots, \mathbf{O}_K\mathbf{H}_K)$, where $\mathrm{SaNN}(\mathbf{X}, \mathbf{B}) = \{\mathbf{H}_k \in \mathbb{R}^{N_k \times \bar{D}_k}\}_{k=0}^K$ are the combined embedding matrices of $k$-simplices from different depths output by SaNN.

*Proof.* The aggregated feature matrix of a $k$-dimensional simplex output by the $t$th branch of SaNN corresponding to $t$-hop neighborhood aggregation is given by (30). We first prove that if the orientations of the input features to SaNN are modified using $\mathcal{O}$, then the orientations of the 1-hop aggregated features are changed by the same $\mathcal{O}$. We then prove, by induction, that if the orientations of the input features are changed using $\mathcal{O}$, then the orientations of the aggregated features of any depth are permuted by the same $\mathcal{O}$.

For the base case, when $t = 1$, the aggregated feature matrix output by 1-hop neighbourhood aggregation when the input is $(\mathcal{O}\mathbf{X}, \mathcal{O}\mathbf{B}\mathcal{O}^{\mathrm{T}})$ is the following

$$\begin{aligned}
\bar{\mathbf{X}}_k^{(1)} &= \Big[(\mathbf{O}_k\mathbf{B}_{k+1}\mathbf{O}_{k+1}^{\mathrm{T}}\mathbf{O}_{k+1}\mathbf{B}_{k+1}^{\mathrm{T}}\mathbf{O}_k^{\mathrm{T}}) \cdot (\mathbf{O}_k\mathbf{X}_k), (\mathbf{O}_k\mathbf{B}_k^{\mathrm{T}}\mathbf{O}_{k-1}^{\mathrm{T}}\mathbf{O}_{k-1}\mathbf{B}_k\mathbf{O}_k^{\mathrm{T}}) \cdot (\mathbf{O}_k\mathbf{X}_k), \\
&\quad (\mathbf{O}_k\mathbf{B}_k^{\mathrm{T}}\mathbf{O}_{k-1}^{\mathrm{T}}) \cdot (\mathbf{O}_{k-1}\mathbf{X}_{k-1}), (\mathbf{O}_k\mathbf{B}_{k+1}\mathbf{O}_{k+1}^{\mathrm{T}}) \cdot (\mathbf{O}_{k+1}\mathbf{X}_{k+1})\Big]^{\mathrm{T}}, \\
&= \Big[\mathbf{O}_k\mathbf{B}_{k+1}\mathbf{B}_{k+1}^{\mathrm{T}}\mathbf{X}_k, \mathbf{O}_k\mathbf{B}_k^{\mathrm{T}}\mathbf{B}_k\mathbf{X}_k, \ \mathbf{O}_k\mathbf{B}_k^{\mathrm{T}}\mathbf{X}_{k-1}, \mathbf{O}_k\mathbf{B}_{k+1}\mathbf{X}_{k+1}\Big]^{\mathrm{T}}, \\
&= \mathbf{O}_k\Big[\mathbf{B}_{k+1}\mathbf{B}_{k+1}^{\mathrm{T}}\mathbf{X}_k, \mathbf{B}_k^{\mathrm{T}}\mathbf{B}_k\mathbf{X}_k, \ \mathbf{B}_k^{\mathrm{T}}\mathbf{X}_{k-1}, \mathbf{B}_{k+1}\mathbf{X}_{k+1}\Big]^{\mathrm{T}}, \\
&= \mathbf{O}_k\mathbf{X}_k^{(1)}
\end{aligned} \tag{33}$$

for $k = 0, \ldots, K$. Second equality is achieved using the fact that $\mathbf{O}_k\mathbf{O}_k^{\mathrm{T}} = \mathbf{I}, \forall k \in 0, \ldots, K$. This shows that when the inputs $\mathbf{X}$ and $\mathbf{B}$ are changed to $\mathcal{O}\mathbf{X}$ and $\mathcal{O}\mathbf{B}\mathcal{O}^{\mathrm{T}}$, the orientations of the 1-hop aggregated features $\mathbf{X}_k^{(1)} \in \mathbb{R}^{N_k \times D_k}\}_{k=0}^K$ are also altered by the same set of matrices.

Let us assume that when the inputs are $\mathcal{O}\mathbf{X}$ and $\mathcal{O}\mathbf{B}\mathcal{O}^{\mathrm{T}}$, the $(t-1)$-hop aggregated features $\{\mathbf{X}_k^{(t-1)} \in \mathbb{R}^{N_k \times D_k}\}_{k=1}^K$ are also altered by the same set of matrices, i.e., $\bar{\mathbf{X}}_k^{(t-1)} = \mathbf{O}_k\mathbf{X}_k^{(t-1)}$. Using the induction hypothesis, the aggregated feature matrix output by $t$-hop neighbourhood aggregation when the input is

$(\mathcal{O}\mathbf{X}, \mathcal{O}\mathbf{B}\mathcal{O}^{\mathrm{T}})$ is the following

$$
\begin{aligned}
\bar{\mathbf{X}}_k^{(t)} &= \Big[(\mathbf{O}_k\mathbf{B}_{k+1}\mathbf{O}_{k+1}^{\mathrm{T}}\mathbf{O}_{k+1}\mathbf{B}_{k+1}^{\mathrm{T}}\mathbf{O}_k^{\mathrm{T}})\cdot(\bar{\mathbf{X}}_k^{(t-1)}), (\mathbf{O}_k\mathbf{B}_k^{\mathrm{T}}\mathbf{O}_{k-1}^{\mathrm{T}}\mathbf{O}_{k-1}\mathbf{B}_k\mathbf{O}_k^{\mathrm{T}})\cdot(\bar{\mathbf{X}}_k^{(t-1)}), \\
&\quad (\mathbf{O}_k\mathbf{B}_k^{\mathrm{T}}\mathbf{O}_{k-1}^{\mathrm{T}})\cdot(\bar{\mathbf{X}}_{k-1}^{(t-1)}), (\mathbf{O}_k\mathbf{B}_{k+1}\mathbf{O}_{k+1}^{\mathrm{T}})\cdot(\bar{\mathbf{X}}_{k+1}^{(t-1)})\Big]^{\mathrm{T}}, \\
&= \Big[(\mathbf{O}_k\mathbf{B}_{k+1}\mathbf{O}_{k+1}^{\mathrm{T}}\mathbf{O}_{k+1}\mathbf{B}_{k+1}^{\mathrm{T}}\mathbf{O}_k^{\mathrm{T}})\cdot(\mathbf{O}_k\mathbf{X}_k^{(t-1)}), (\mathbf{O}_k\mathbf{B}_k^{\mathrm{T}}\mathbf{O}_{k-1}^{\mathrm{T}}\mathbf{O}_{k-1}\mathbf{B}_k\mathbf{O}_k^{\mathrm{T}})\cdot(\mathbf{O}_k\mathbf{X}_k^{(t-1)}), \\
&\quad (\mathbf{O}_k\mathbf{B}_k^{\mathrm{T}}\mathbf{O}_{k-1}^{\mathrm{T}})\cdot(\mathbf{O}_{k-1}\mathbf{X}_{k-1}^{(t-1)}), (\mathbf{O}_k\mathbf{B}_{k+1}\mathbf{O}_{k+1}^{\mathrm{T}})\cdot(\mathbf{O}_{k+1}\mathbf{X}_{k+1}^{(t-1)})\Big]^{\mathrm{T}}, \\
&= \Big[\mathbf{O}_k\mathbf{B}_{k+1}\mathbf{B}_{k+1}^{\mathrm{T}}\mathbf{X}_k^{(t-1)}, \mathbf{O}_k\mathbf{B}_k^{\mathrm{T}}\mathbf{B}_k\mathbf{X}_k^{(t-1)}, \; \mathbf{O}_k\mathbf{B}_k^{\mathrm{T}}\mathbf{X}_{k-1}^{(t-1)}, \mathbf{O}_k\mathbf{B}_{k+1}\mathbf{X}_{k+1}^{(t-1)}\Big]^{\mathrm{T}}, \\
&= \mathbf{O}_k\Big[\mathbf{B}_{k+1}\mathbf{B}_{k+1}^{\mathrm{T}}\mathbf{X}_k^{(t-1)}, \mathbf{B}_k^{\mathrm{T}}\mathbf{B}_k\mathbf{X}_k^{(t-1)}, \; \mathbf{B}_k^{\mathrm{T}}\mathbf{X}_{k-1}^{(t-1)}, \mathbf{B}_{k+1}\mathbf{X}_{k+1}^{(t-1)}\Big]^{\mathrm{T}}, \\
&= \mathbf{O}_k\mathbf{X}_k^{(t)}, \quad (34)
\end{aligned}
$$

where the second equality follows from the induction hypothesis. This proves that the neighborhood aggregation in SaNN is orientation equivariant.

Recall that we choose the transformation functions $g_k^{(t)}(\cdot)$ to be MLPs and the functions that combine embeddings from different hops, i.e., $\Theta_k(\cdot)$, to be concatenation. Without loss of generality, assume that the feature transformation functions are two-layered MLPs. The final output embedding matrices of $k$-dimensional simplices $\bar{\mathbf{H}}_k$ can be written as

$$
\begin{aligned}
\bar{\mathbf{H}}_k &= \Big[\eta_{k,1}^{(0)}\big(\eta_{k,0}^{(0)}\big(\bar{\mathbf{X}}_k^{(0)}\mathbf{W}_{k,0}^{(0)}\big)\mathbf{W}_{k,1}^{(0)}\big), \ldots, \eta_{k,1}^{(T)}\big(\eta_{k,0}^{(T)}\big(\bar{\mathbf{X}}_k^{(T)}\mathbf{W}_{k,0}^{(T)}\big)\mathbf{W}_{k,1}^{(T)}\big)\Big]^{\mathrm{T}} \\
&= \Big[\eta_{k,1}^{(0)}\big(\eta_{k,0}^{(0)}\big(\mathbf{O}_k\mathbf{X}_k^{(0)}\mathbf{W}_{k,0}^{(0)}\big)\mathbf{W}_{k,1}^{(0)}\big), \ldots, \eta_{k,1}^{(T)}\big(\eta_{k,0}^{(T)}\big(\mathbf{O}_k\mathbf{X}_k^{(T)}\mathbf{W}_{k,0}^{(T)}\big)\mathbf{W}_{k,1}^{(T)}\big)\Big]^{\mathrm{T}}, \quad (35)
\end{aligned}
$$

for some learnable weight matrices $\mathbf{W}_{k,i}^{(t)}$ and activation functions $\eta_{k,i}^{(t)}, \forall k = 0, \ldots, K,\, t = 0, \ldots, T,\, i = 0, 1$.

The linear transformation part of an MLP is clearly orientation equivariant. The non-linear activation part of an MLP is orientation equivariant only if it is odd. Specifically, if an activation function $\eta$ is odd, then $\eta(\mathbf{OA}) = \mathbf{O}\eta(\mathbf{A})$ for all matrices $\mathbf{A}$ of appropriate dimension, and any diagonal matrix $\mathbf{O}$ with diagonal entries as $1$ or $-1$. Therefore, $\bar{\mathbf{H}}_k$ simplifies to

$$
\begin{aligned}
\bar{\mathbf{H}}_k &= \Big[\eta_{k,1}^{(0)}\big(\mathbf{O}_k\big(\eta_{k,0}^{(0)}\big(\mathbf{X}_k^{(0)}\mathbf{W}_{k,0}^{(0)}\big)\big)\mathbf{W}_{k,1}^{(0)}\big), \ldots, \eta_{k,1}^{(T)}\big(\big(\eta_{k,0}^{(T)}\big(\mathbf{X}_k^{(T)}\mathbf{W}_{k,0}^{(T)}\big)\big)\mathbf{W}_{k,1}^{(T)}\big)\Big]^{\mathrm{T}}, \\
&= \Big[\mathbf{O}_k\big(\eta_{k,1}^{(0)}\big(\eta_{k,0}^{(0)}\big(\mathbf{X}_k^{(0)}\mathbf{W}_{k,0}^{(0)}\big)\big)\mathbf{W}_{k,1}^{(0)}\big), \ldots, \mathbf{O}_k\big(\eta_{k,1}^{(T)}\big(\eta_{k,0}^{(T)}\big(\mathbf{X}_k^{(T)}\mathbf{W}_{k,0}^{(T)}\big)\big)\mathbf{W}_{k,1}^{(T)}\big)\Big]^{\mathrm{T}}, \\
&= \mathbf{O}_k\Big[\big(\eta_{k,1}^{(0)}\big(\eta_{k,0}^{(0)}\big(\mathbf{X}_k^{(0)}\mathbf{W}_{k,0}^{(0)}\big)\big)\mathbf{W}_{k,1}^{(0)}\big), \ldots, \big(\eta_{k,1}^{(T)}\big(\eta_{k,0}^{(T)}\big(\mathbf{X}_k^{(T)}\mathbf{W}_{k,0}^{(T)}\big)\big)\mathbf{W}_{k,1}^{(T)}\big)\Big]^{\mathrm{T}}, \\
&= \mathbf{O}_k\mathbf{H}_k. \quad (36)
\end{aligned}
$$

This proves orientation equivariance of SaNN. $\qquad\square$

## H   EXPERIMENTAL SETUP AND DATASETS

The experimental setup for the three tasks considered is discussed here. For a fair computational comparison, the same predictive models, and hyperparameters, namely depth ($T$), order ($K$), and hidden dimensions, were utilized for all the baseline SNN models.

### H.1   TRAJECTORY PREDICTION

For each dataset in trajectory prediction experiments, we assign to each trajectory an attribute of dimension the same as the number of edges in the entire simplicial complex corresponding to that dataset. The $i$th entry of the trajectory attribute vector is $0$ if the $i$th edge is not a part of the trajectory, is $1$ if it is a part of the trajectory and has an orientation same as that of the trajectory, and is $-1$ if it is a part of the trajectory but has an opposite orientation relative to that of the trajectory. We then aggregate the features of upper and lower neighborhood edges from different hops as discussed in Section V in the manuscript using normalized incidence matrices. We perform a $80\% - 20\%$ split of the data for training and testing purposes. We use 5-fold cross-validation for

evaluating the performance of each of the deep models considered including `SaNN`. The model is trained three times on data corresponding to each of the 5 folds to eliminate any weight initialization biases.

To use the edge embeddings learnt using `SaNN` to perform trajectory prediction, we map the embeddings from the vector space of edge embeddings to the vector space of node embeddings by multiplying them by the incidence matrix $\mathbf{B}_1^{\mathrm{T}} \in \mathbb{R}^{N_1 \times N_0}$. We consider the node embeddings of only those nodes that have a possibility of being the next in the trajectory, which are the neighboring nodes of the last node in a trajcetory. For a trajectory $t$, which ends at node $i$, we use a masking function to extract the embeddings corresponding to the neighboring nodes of the node $i$. We do this for node embeddings from different depths and combine them using an injective readout function, specifically, concatenation. We then give the combined node embeddings as input to a decoder MLP which maps the node embeddings to a vector space of dimension same as the maximum number of neighboring nodes of the last node in any trajectory in the dataset. The last layer of the decoder is a softmax activation function, the output of which for a trajectory is a vector containing the probabilities with which the neighboring nodes of the trajectory's last node become the successor node of that trajectory. For all the datasets, we train `SaNN` in a supervised manner using the cross-entropy loss function. We implement early stopping while training, i.e., terminate the training process if there is no significant gain in the validation accuracy after a certain number of training epochs. It is established that ensuring that the models are orientation equivariant enhances their trajectory prediction performance (Roddenberry et al., 2021). We, therefore, use tanh activation function in all the layers of `SaNN` and all the other existing SNN variants [c.f. Section 4.2].

Details of the datasets used in trajectory prediction experiments, namely, the number of nodes, edges, and triangles, in the simplicial complexes, number of trajectories in the datasets and the average length of the trajectories, are summarized in Table 4.

Table 4: Trajectory prediction dataset details.

| DATASET | NODES | EDGES | TRIANGLES | PATHS | AVG. PATH LENGTH |
|---------|-------|-------|-----------|-------|------------------|
| OCEAN | 133 | 320 | 186 | 160 | 5.73 |
| SYNTHETIC | 400 | 1001 | 649 | 800 | 8.20 |
| PLANAR | 500 | 2857 | 5842 | 3306 | 2.96 |
| MESH | 400 | 1482 | 1444 | 3311 | 3.18 |

The hyperparameters associated with trajectory prediction experiments, namely, batch size, number of branches, given by $(T + 1)$, used in the computation of features of edges from different hops, hidden dimensions of edge representations in the MLP for transforming edge features (denoted by T) and the decoder MLP (denoted by D), learning rate, $L_2$ regularization parameters, are summarized in Table 5.

Table 5: Hyperparameters of `SaNN` set for trajectory prediction experiments.

| DATASET | BATCH SIZE | BRANCHES $(T+1)$ | HIDDEN DIMENSIONS | LEARNING RATE | $L_2$ REGULARIZATION |
|---------|-----------|------------------|-------------------|---------------|----------------------|
| OCEAN | 32 | 3 | T:{320,320,320,320} D:{133,133,133,6} | $10^{-3}$ | $10^{-4}$ |
| SYNTHETIC | 8 | 3 | T:{1001,1001,1001,1001} D:{400,400,400,13} | $10^{-5}$ | $10^{-4}$ |
| PLANAR | 8 | 3 | T:{2857,2857,2857,2857} D:{500,500,500,17} | $10^{-5}$ | $10^{-4}$ |
| MESH | 16 | 3 | T:{1482,1482,1482} D:{400,400,400,7} | $10^{-5}$ | $10^{-4}$ |

## H.2 SIMPLICIAL CLOSURE PREDICTION

The list of all the open triangles in the simplicial complex formed till $80\%$ time instances in the datasets and the corresponding labels of whether they get closed by the time the entire simplicial complex is formed will form the data. We perform a $60\% - 20\% - 20\%$ split of the data for training, validation, and testing. For all simplicial closure prediction experiments, we consider the initial features on simplices as the sum of the features aggregated from their upper and lower adjacent simplices.

To perform simplicial closure prediction for triangles, we concatenate the embeddings of the constituent nodes and edges in the open triangles in the training set learnt using `SaNN`. The concatenated embeddings are given as input to a decoder MLP with a sigmoid activation function in the last layer, the output of which is the probability of the open triangles forming a simplex. For all the datasets, we train `SaNN` in a supervised manner using the binary cross-entropy loss function. We implement early stopping while training. Since the datasets are heavily skewed with many negative examples (open triangles that never get closed) than positive examples (open triangles that get closed), we perform oversampling of the positive examples in each training batch and use a weight for positive examples in the binary cross-entropy loss function.

Details of the datasets used in simplicial closure prediction experiments, namely, the number of nodes, edges, and triangles, in the simplicial complexes, and the number of open triangles in the training data, are summarized in Table 6.

Table 6: Simplicial closure prediction dataset details.

| DATASET | 0-SIMPLICES | 1-SIMPLICES | 2-SIMPLICES | OPEN TRIANGLES |
|---|---|---|---|---|
| EMAIL-ENRON | 142 | 1655 | 6095 | 2574 |
| CONTACT-HIGH-SCHOOL | 327 | 5818 | 2370 | 25056 |
| CONTACT-PRIMARY-SCHOOL | 242 | 8317 | 5139 | 80446 |
| NDC-CLASSES | 1161 | 6222 | 31477 | 7439 |
| TAGS-MATH-SX | 1629 | 91685 | 249548 | 1960269 |

The hyperparameters associated with simplicial closure prediction experiments, namely, batch size, number of branches $(T + 1)$ used in the computation of features of each of the $K = 2$ different dimensional simplices, namely, nodes and edges, hidden dimensions of node and edge representations in the MLP for transforming edge features (denoted by T) and the decoder MLP (denoted by D), learning rate, $L_2$ regularization parameters, and the weight assigned to positive examples in the cross-entropy loss function, are summarized in Table 7.

Table 7: Hyperparameters of `SaNN` set for simplicial closure prediction experiments.

| DATASET | BATCH SIZE | BRANCHES $(K \times (T + 1))$ | HIDDEN DIMENSIONS | LEARNING RATE | $L_2$ REGULARIZATION | WEIGHT OF POSITIVE EXAMPLES |
|---|---|---|---|---|---|---|
| EMAIL-ENRON | 32 | $2 \times 3$ | T:{64,64,64} D:{64,1} | $10^{-2}$ | $10^{-3}$ | 3 |
| CONTACT-HIGH-SCHOOL | 32 | $2 \times 3$ | T:{32,32} D:{64,1} | $10^{-2}$ | $10^{-3}$ | 3 |
| CONTACT-PRIMARY-SCHOOL | 32 | $2 \times 3$ | T:{32,32,32,32} D:{64,64,64,1} | $10^{-2}$ | $10^{-3}$ | 3 |
| NDC-CLASSES | 32 | $2 \times 3$ | T:{32,32,32} D:{64,64,1} | $10^{-3}$ | $10^{-3}$ | 3 |
| TAGS-MATH-SX | 32 | $2 \times 3$ | T:{32,32} D:{64,64,64,1} | $10^{-2}$ | $10^{-3}$ | 3 |

## H.3 GRAPH CLASSIFICATION

We lift the graphs in all the graph classification datasets to simplicial complexes by considering 3-cliques as 2-simplices. For all the graph classification experiments, we use input node attributes as suggested by Errica et al. (2020) and input attributes of higher-order simplices as all-one vectors of the same dimension as the respective node attributes. We use the predefined data splits that are used in Errica et al. (2020) for performing 10-fold cross-validation. Within data corresponding to each fold, we holdout 10% for validation and use the rest for training. The model is trained three times on data corresponding to each of the 10 folds to eliminate any weight initialization biases. Details of the datasets used in graph classification experiments, namely, the number of graphs in the dataset, the average number of nodes, edges, and triangles in the graphs, and the dimension of the initial attributes of simplices, are summarized in Table 8. To perform graph classification using the embeddings learnt using `SaNN`, we combine embeddings of the constituent simplices of all orders in the clique-lifted graphs. Since the number of $k$-simplices in a training example (a clique-lifted graph) is large for $k = 0, 1, \ldots, K$, concatenating embeddings of all the $k$-simplices in a training example will result in very high-dimensional embeddings. Therefore, we propose to use summation as the readout function at $k$-simplices level. We, finally, use concatenation as the global-level readout function. Specifically, we take a summation of all the node embeddings, edge embeddings, and triangle embeddings in a graph and concatenate the node-level, edge-level, and triangle-level embeddings of the simplicial complex to get one global representation for the simplicial complex. We prove that using summation as the $k$-simplices level readout function along with

concatenation as the global level readout function preserves injectivity in Appendix I. The global embeddings of graphs are given as input to a decoder MLP with either a sigmoid activation function or a softmax activation function as the final layer depending on whether the dataset has two or more classes. For all the datasets, we train SaNN in a supervised manner using the cross-entropy loss function, which takes the output of the decoder and the true graph labels as the input. We implement early stopping while training.

Table 8: Graph classification dataset details.

| DATASET | GRAPHS | CLASSES | AVG. NODES | AVG. EGDES | AVG. TRIANGLES | INPUT FEAT. DIM. |
|---|---|---|---|---|---|---|
| PROTEINS | 1113 | 2 | 39.06 | 72.82 | 27.40 | 3 |
| NCI1 | 4110 | 2 | 29.87 | 32.3 | 0.05 | 37 |
| IMDB-B | 1000 | 2 | 19.77 | 96.53 | 391.99 | 1 |
| IMDB-M | 1500 | 3 | 13.00 | 65.94 | 305.9 | 1 |
| REDDIT-B | 2000 | 2 | 429.63 | 497.75 | 24.83 | 1 |
| REDDIT-M | 5000 | 5 | 508.52 | 594.87 | 21.77 | 1 |

The hyperparameters used in graph classification experiments, namely, batch size, number of branches, given by $(T + 1)$, used in the computation of the features of each of the $K = 3$ different dimensional simplices, namely, nodes, edges, and triangles, hidden dimensions of edge representations in the MLP for transforming edge features (denoted by T) and the decoder MLP (denoted by D), learning rate, $L_2$ regularization parameters, are summarized in Table 9.

Table 9: Hyperparameters of SaNN set for graph classification experiments.

| DATASET | BATCH SIZE | BRANCHES $(K \times (T + 1))$ | HIDDEN DIMENSIONS | LEARNING RATE | $L_2$ REGULARIZATION |
|---|---|---|---|---|---|
| PROTEINS | 64 | $3 \times 2$ | {64,64,64} {64,64,64,1} | $10^{-3}$ | $10^{-4}$ |
| NCI1 | 64 | $3 \times 2$ | {64,64,64} {64,64,64,1} | $10^{-3}$ | $10^{-4}$ |
| IMDB-B | 64 | $3 \times 2$ | {32,32,32} {32,32,32,1} | $10^{-3}$ | $10^{-4}$ |
| IMDB-M | 64 | $3 \times 2$ | {16,16,16} {16,16,3} | $10^{-4}$ | $10^{-4}$ |
| REDDIT-B | 64 | $3 \times 2$ | {32,32,32} {32,32,1} | $10^{-3}$ | $10^{-3}$ |
| REDDIT-M | 64 | $3 \times 2$ | {64,64,64} {64,64,5} | $5 \times 10^{-4}$ | $10^{-4}$ |

# I  SUMMATION AS $k$-SIMPLICES-LEVEL READOUT FUNCTION AND CONCATENATION AS GLOBAL READOUT FUNCTIONS ARE INJECTIVE

Let us denote the multiset of the aggregated features at branch $t$ of the $k$-simplices in a training example (here, a clique-lifted graph) $\mathcal{T}_k$ by $\mathcal{A}_k^{(t)} = \left\{\!\!\left\{ \mathbf{X}_k^{(t)}[\tau], \forall \tau \in \mathcal{T}_k \right\}\!\!\right\}$. From Lemma 5 in Leskovec & Jegelka (2019), there exists functions $g_k^{(t)}(\cdot)$, $\forall t \in 0, \dots, T$, so that $\sum_{a \in \mathcal{A}_k^{(t)}} g_k^{(t)}(a)$ are unique for each multiset $\mathcal{A}_k^{(t)}$ computed from the training set. Since concatenation is injective, this implies that there exist functions $g_k^{(t)}(\cdot)$, $\forall t \in 0, \dots, T$, such that the combined embeddings of $k$-simplices in the training example from different hops

$$\mathbf{e}_k = \left[ \sum_{a \in \mathcal{A}_k^{(0)}} g_k^{(0)}(a), \sum_{a \in \mathcal{A}_k^{(1)}} g_k^{(1)}(a), \dots, \sum_{a \in \mathcal{A}_k^{(T)}} g_k^{(T)}(a) \right]^{\mathrm{T}}$$

(37)

is unique. Given the universal approximation ability of MLPs and our choice of using MLPs as the transformation functions, the injectivity of SaNN is preserved.

Table 10: Trajectory prediction accuracies of ablation experiments to study the effect of features from different depths. The first and the second best performances are highlighted in red and blue, respectively. The values within parentheses are the average per epoch run-time values (in seconds). The best run-time values are highlighted in bold.

| Dataset | Ocean | Synthetic | Planar | Mesh |
|---|---|---|---|---|
| SaNN (1-hop) | $34.0 \pm 0.6$ **(0.1)** | $50.3 \pm 1.2$ **(2.0)** | $96.9 \pm 1.9$ **(17.6)** | $98.1 \pm 1.9$ **(4.1)** |
| SaNN (2-hop) | $34.0 \pm 0.8$ **(0.1)** | $53.8 \pm 1.1$ (2.1) | $96.8 \pm 1.8$ (17.9) | $95.5 \pm 2.0$ (4.9) |
| SaNN (3-hop) | $33.5 \pm 0.7$ **(0.1)** | $51.3 \pm 1.2$ (2.1) | $95.5 \pm 2.0$ (19.2) | $94.3 \pm 2.0$ (4.9) |
| SaNN (T=2) | $35.0 \pm 0.7$ **(0.1)** | $54.9 \pm 1.1$ (2.5) | $97.0 \pm 1.9$ (23.8) | $98.3 \pm 2.0$ (8.1) |
| SaNN (T=3) | $34.1 \pm 0.7$ **(0.1)** | $53.9 \pm 1.1$ (2.5) | $97.3 \pm 1.9$ (26.8) | $98.3 \pm 2.1$ (8.1) |

Notice that concatenating the readouts from different branches of $k$-simplices is the same as concatenating the embeddings from different branches first to get embeddings for all the $k$-simplices and then computing the readout:

$$
\mathbf{e}_k = \left[ \sum_{a \in \mathcal{A}_k^{(0)}} g_k^{(0)}(a), \sum_{a \in \mathcal{A}_k^{(1)}} g_k^{(1)}(a), \ldots, \sum_{a \in \mathcal{A}_k^{(T)}} g_k^{(T)}(a) \right]^{\mathrm{T}}
$$

$$
= \left[ \sum_{\tau \in \mathcal{T}_k} \mathbf{S}_k^{(0)}[\tau], \sum_{\tau \in \mathcal{T}_k} \mathbf{S}_k^{(1)}[\tau], \ldots, \sum_{\tau \in \mathcal{T}_k} \mathbf{S}_k^{(T)}[\tau] \right]^{\mathrm{T}}
$$

$$
= \sum_{\tau \in \mathcal{T}_k} \left[ \mathbf{S}_k^{(0)}[\tau], \mathbf{S}_k^{(1)}[\tau], \ldots, \mathbf{S}_k^{(T)}[\tau] \right]^{\mathrm{T}}. \tag{38}
$$

## J  ABLATION STUDY

We perform ablation experiments to study the effect of features from the different components of the SaNN model. Specifically, we perform the following two experiments: (i) we study the effect of precomputed features aggregated from different depths of neighborhood, i.e., the effect of features for different $t$, and (ii) we study the effect of features of simplices of different orders, i.e., the effect of features for different $k$, for the tasks described in Section 5.

Table 11: Relative AUC-PR values of ablation experiments to study the effect of features from different depths for simplicial closure prediction. The first and the second best performances are highlighted in red and blue, respectively. The values within parentheses are the average per epoch run-time values (in seconds). The best run-time values are highlighted in bold.

| DATASET | ENRON | HIGH-SCHOOL | PRIMARY-SCHOOL | NDC-CLASSES | MATH-SX |
|---|---|---|---|---|---|
| SaNN (1-HOP)/RB | $15.18 \pm 0.1$ **(2)** | $29.86 \pm 0.0$ **(28)** | $30.19 \pm 0.0$ **(133)** | $1.82 \pm 0.1$ **(7)** | $6.33 \pm 0.0$ **(18396)** |
| SaNN (2-HOP)/RB | $14.88 \pm 0.1$ **(2)** | $29.93 \pm 0.0$ (96) | $33.18 \pm 0.0$ (138) | $2.53 \pm 0.1$ (8) | $5.83 \pm 0.0$ (32612) |
| SaNN (3-HOP)/RB | $14.78 \pm 0.0$ **(2)** | $26.61 \pm 0.1$ (107) | $30.66 \pm 0.0$ (594) | $0.95 \pm 0.1$ (9) | $4.99 \pm 0.1$ (45716) |
| SaNN (T=2)/RB | $15.10 \pm 0.0$ (3) | $30.22 \pm 0.0$ (112) | $32.89 \pm 0.0$ (916) | $2.79 \pm 0.0$ (13) | $6.88 \pm 0.0$ (52883) |
| SaNN (T=3)/RB | $15.45 \pm 0.0$ (3) | $26.63 \pm 0.0$ (203) | $32.28 \pm 0.0$ (932) | $1.75 \pm 0.0$ (22) | $5.65 \pm 0.1$ (78323) |

Table 12: Graph classification accuracies of ablation experiments to study the effect of features from different depths. The first and the second best performances are highlighted in red and blue, respectively. The values within parentheses are the average per epoch run-time values (in seconds). The best run-time values are highlighted in bold.

| DATASET | PROTEINS | NCI1 | IMDB-B | REDDIT-B | REDDIT-M |
|---|---|---|---|---|---|
| SaNN (1−HOP) | **75.0 ± 2.2** **(0.3)** | 73.0 ± 2.0 **(55)** | **70.3 ± 2.1** **(7)** | **91.8 ± 2.7** **(39)** | **53.0 ± 1.6** **(102)** |
| SaNN (2−HOP) | 74.9 ± 2.1 (0.3) | **74.1 ± 2.2** (57) | 70.2 ± 2.0 (8) | 90.3 ± 2.7 (40) | 52.4 ± 1.5 (103) |
| SaNN (T=2) | **77.6 ± 2.2** (0.4) | **74.9 ± 2.2** (58) | **72.7 ± 2.1** (8) | **91.7 ± 2.7** (45) | **54.1 ± 1.6** (104) |

Table 13: Relative AUC-PR values of ablation experiments to study the effect of features of simplices of different orders for simplicial-closure prediction. The first and the second best performances are highlighted in red and blue, respectively. The values within parentheses are the average per epoch run-time values (in seconds).

| DATASET | ENRON | HIGH-SCHOOL | PRIMARY-SCHOOL | NDC-CLASSES | MATH-SX |
|---|---|---|---|---|---|
| SaNN (0−SIMPLICES) | 11.14 ± 0.0 **(2)** | 16.61 ± 0.0 (85) | 4.19 ± 0.0 **(718)** | **2.62 ± 0.0** (12) | 4.96 ± 0.0 **(33618)** |
| SaNN (1−SIMPLICES) | **13.03 ± 0.0** **(2)** | **27.11 ± 0.0** **(84)** | **30.28 ± 0.0** (719) | 2.52 ± 0.1 **(10)** | **6.18 ± 0.0** (42422) |
| SaNN (K=1)/RB | **15.45 ± 0.0** (3) | **30.22 ± 0.0** (112) | **32.89 ± 0.0** (916) | **2.79 ± 0.0** (13) | **6.88 ± 0.0** (52883) |

## J.1 EFFECT OF FEATURES FROM DIFFERENT DEPTHS

We study the influence that the features aggregated from the 1-hop neighborhood, 2-hop neighborhood, and 3-hop neighborhood have on the prediction accuracies for trajectory classification and AUC-PR values for simplicial closure prediction. Since the graphs in graph classification datasets are very small, we restrict the ablation experiments for graph classification to the features aggregated from the 1-hop and 2-hop neighborhoods. We also study the effect of combining the features from different depths as proposed in the paper [cf. Fig. 2] relative to using features aggregated from a specific depth of neighborhood. The results of ablation experiments for trajectory prediction, simplicial closure prediction, and graph classification are reported in Tables 10, 11, and 12, respectively. Experiments involving features only from depth $t$ are denoted by SaNN (t−hop) and those involving features aggregated from $0, \ldots, t$-hop neighborhood are denoted by SaNN (T=t).

We observe that the proposed model (SaNN (T=2(or 3))) as given in Section 3, which considers the combination of features from different depths, achieves better performance than using features from a specific neighborhood depth (SaNN (t−hop), for $t = 1, 2, 3$) for all the three tasks. Since the model proposed in Section 3, denoted by SaNN (T=2(or 3)) (and not (SaNN (t−hop), for $t = 1, 2, 3$)), is what is proved to be as powerful as MPSN, we expect its embeddings to be as expressive as those from MPSN, and therefore, perform downstream tasks on par with MPSN.

We observe that 1-hop information is usually the most informative, implying that local information is crucial for all three tasks. The 3-hop information seems not so relevant for these tasks. For the cases when the combination of information from different depths is considered, SaNN (T=2) is seen to perform better than SaNN (T=3) possibly due to the inclusion of the irrelevant non-local features in the form of 3-hop features. This implies that if an even faster implementation is desired at the cost of a slight degradation of performance, SaNN (1−hop) can be used since it achieves comparable performance as SaNN (T=2 (or 3)) in an even smaller run-time. However, for graph classification, we observe that since the graph sizes are small, the run-time difference between SaNN (t−hop), for $t = 1, 2$, and SaNN (T=2) is not prominent. Therefore, it is always better to perform graph classification using SaNN (T=2) than SaNN (t−hop), for $t = 1, 2$.

## J.2 EFFECT OF FEATURES OF SIMPLICES OF DIFFERENT ORDERS

We study the influence that simplices of different orders have on the AUC-PR values for simplicial closure prediction and classification accuracies for graph classification. Since we focus on the simplicial closure of open triangles, the ablation experiments involve features of only 0-simplices and 1-simplices. In the ablation experiments for graph classification, we consider features of 0, 1, 2, and 3 simplices. Since features on only

Table 14: Graph classification accuracies of ablation experiments to study the effect of features of simplices of different orders. The first and the second best performances are highlighted in red and blue, respectively. The absence of simplices of a particular order in a dataset is indicated by - in the corresponding column.

| DATASET | PROTEINS | NCI1 | IMDB-B | REDDIT-B | REDDIT-M |
|---|---|---|---|---|---|
| SaNN (0-SIMPLICES) | $76.0 \pm 2.3$ (0.2) | $\mathbf{73.5 \pm 2.2}$ (32) | $\mathbf{71.2 \pm 2.1}$ (7) | $88.6 \pm 2.6$ (41) | $\mathbf{53.6 \pm 1.5}$ (91) |
| SaNN (1-SIMPLICES) | $59.3 \pm 2.3$ (0.2) | $66.2 \pm 2.3$ (35) | $70.6 \pm 2.2$ (8) | $\mathbf{88.7 \pm 2.6}$ (42) | $38.1 \pm 1.6$ (84) |
| SaNN (2-SIMPLICES) | $73.4 \pm 2.1$ (0.3) | $-$ | $70.8 \pm 2.1$ (8) | $75.8 \pm 2.8$ (37) | $43.0 \pm 1.6$ (89) |
| SaNN (3-SIMPLICES) | $60.0 \pm 2.3$ (0.3) | $-$ | $59.1 \pm 2.3$ (8) | $-$ | $-$ |
| SaNN (K=2) | $\mathbf{76.2 \pm 2.2}$ (0.3) | $\mathbf{74.9 \pm 2.2}$ (58) | $\mathbf{72.7 \pm 2.1}$ (8) | $\mathbf{91.7 \pm 2.7}$ (45) | $\mathbf{54.1 \pm 1.6}$ (104) |
| SaNN (K=3) | $\mathbf{77.6 \pm 2.2}$ (0.4) | $-$ | $68.2 \pm 2.1$ (9) | $-$ | $-$ |

1-simplices are considered for trajectory prediction, we do not perform such an ablation experiment for the trajectory prediction application. The results of ablation experiments for simplicial closure prediction and graph classification are reported in Tables 13 and 14, respectively. Experiments involving features corresponding to only $k$-simplices are denoted by SaNN (k-simplices) and those involving features of $0, \ldots, k$-simplices are denoted by SaNN (K=k).

It is evident from Tables 13 and 14 that combining the transformed features from simplices of different orders results in a much better performance than using only nodes, edges, triangles, or tetrahedrons to the tasks. Specifically, for simplicial closure prediction, we observe that the constituent edges of open triangles carry the most crucial information about whether a simplex will be formed. This agrees with the observation made in Benson et al. (2018), which states that the tie strengths of the edges in an open triangle positively impact its simplicial closure probability. More specifically, how often the constituent pairs of nodes in an open triangle appeared together in a simplex is observed to affect how probable the three nodes are to form a simplex. This explains the superior performance of SaNN (1-simplices) as it captures the interactions of the edges with the other higher-order simplices. Since SaNN (0-simplices) misses capturing this information, its performance is relatively poor. For graph classification, we observe that classification performed using simplices of higher orders like triangles and tetrahedrons yields poor results. This is possibly because of the fact that the graphs are very small and as a result, contain very few simplices of higher orders. The very few simplices present in the graphs are not sufficient to capture the structure of the graphs and distinguish them. Furthermore, we observe that a combination of embeddings of simplices of different orders always results in a better performance than that obtained using simplices of only one order. The result that SaNN (K=2 (or 3)) performs better than SaNN (0-simplices) agrees with the theory, where we prove that SaNN (K=2 (or 3)) is more powerful than the WL test.

