# OpenReview forum: "SaNN: Simple Yet Powerful Simplicial-aware Neural Networks"
_ICLR.cc/2024/Conference — ICLR 2024 spotlight_

### Official Review · Reviewer_4crX · 2023-10-30

**Soundness:** 3 good
**Presentation:** 3 good
**Contribution:** 3 good
**Rating:** 6
**Confidence:** 4

**Summary:**

This paper considers the design of neural networks for simplicial complexes, which are more general combinatorial structures than graphs, but less general than hypergraphs. The authors propose to use multihop aggregation schemes to build an architecture that is more expressive than the simplicial Weisfeiler-Lehman isomorphism test, while satisfying useful invariance, equivariance, and expressivity properties. They also demonstrate the efficientcy of their proposed method, and its performance for a few different tasks in simplicial data processing.

**Strengths:**

1. For the most part, this paper is well-written, and is easy to digest for someone who is familiar with graph neural networks. I don't think the intended audience of this paper includes someone not familiar with GNNs, but this is fine in my opinion.

2. The proposed method is demonstrated to be quite efficient in comparison to existing ones, with similar performance as well.

**Weaknesses:**

1. Certain definitions regarding the types of operators and features are not laid out clearly enough, which leads to ambiguity in the paper on a technical level. As noted in the list of questions and suggestions, the claimed properties of the proposed models are not clearly true, possibly due to this misunderstanding.

**Questions:**

My most important concern is summarized in point 1 -- in particular, the ambiguities around orientation equivariance and the use of oriented operators built from the incidence matrices are what cause me to suggest this paper be rejected. If the authors are to focus on either of the two points in order to change my mind on this paper, it should be the first one.

1. There are some details missing regarding the type of data being handled. In particular, the incidence matrices are not defined in a way sufficient for the discussion following in the paper. Normally, the incidence matrices have values of +-1 depending on a chosen reference orientation (usually given by some ordering of the nodes). Coupled to this, the signs of the features on the simplices are determined relative to the same reference orientation -- this gives meaning to the notion of orientation equivariance. Without discussing these things, orientation equivariance is not a meaningful concept within the context of the paper.

a. This calls into question the validity of the example in Section 4.1. You say that all simplices are given a feature value given by some scalar $a$ -- yet, the matrices acting on these feature vectors/matrices have an orientation associated to them. It seems as if you are using an *oriented operator* to act on *unoriented features*. Property 1 in this example is thus difficult to claim, as the property of orientation equivariance is one describing the action of *oriented operators* acting on *oriented features*, and how the choice of orientation to begin with is irrelevant to the computation.

b. Furthermore, this problem yields a comparison for isomorphism testing incorrect, as the erroneous imposition of differently-oriented features relative to the chosen orientations could be used by SaNN to yield a "false negative," i.e., saying that two isomorphic complexes are different.

c. A more minor comment in this direction comes from the **Insights** section of Section 5.1. It is not correct to say that "the superior performance of SaNN also proves the orientation equivariance of SaNN experimentally." Orientation equivariance is a simple mathematical property, and does not guarantee good performance, nor are all performant architectures on a given dataset orientation equivariant. These properties are possibly linked, but the claim that one proves the other in some way is not justified.

d. Moreover, based on my reading of the appendix, many of their experimental setups for tasks other than trajectory prediction use "unoriented data" by simply assigning scalar values to high-order simplices, which is again incompatible with the use of oriented operators. Perhaps something in the implementation of SaNN in these examples does not use oriented operators such as the incidence matrices, but this is not clear to me.

Please either justify, clarify, or revise the paper's discussion regarding orientation equivariance.

2. Related to the above point, the claims in Section 4.2 seem reasonable at first glance, but are not explained well enough. Permutation equivariance is easily seen to hold, so is not much of a concern. Orientation equivariance is subject to the problems noted above, so more clarification on the type of simplicial features and relevant operators needs to be made. That is not to say that the result proved in the appendix is wrong, but it needs to be clarified in order to be understood in a way that acts on oriented features. Simplicial awareness is more subtle than the other two, based on the definition from (Roddenberry et. al., 2021). For instance, some of the existing convolutional-type SNNs in the literature fail to satisfy simplicial awareness if they are implemented without nonlinear activation functions, due to the fact that the square of the (co)boundary operator is the zero operator. Perhaps it is the case that the assumptions of Theorem 4.2 are sufficient to exclude such methods, but a clearer connection is needed. It would be very helpful for the authors to briefly survey some of the methods they compare to, and clarify whether Theorems 4.1 and 4.2 apply or don't apply to them.

---

Thank you for addressing my questions -- I have raised my suggested score.

---

> ### Author Response · Authors · 2023-11-20
>
> We sincerely appreciate your positive feedback and constructive remarks on our paper. We acknowledge that the unclear definitions may have led to some confusion regarding the properties of the proposed models. Upon reflection, we agree that the orientations of signals and simplices were not sufficiently defined, potentially leading to misunderstandings. We have made revisions to the paper, as detailed below, to enhance clarity and readability.

---

> ### Author Response · Authors · 2023-11-20
> **Clarifying the Orientations of Simplices and Features**
>
> We appreciate your feedback and recognize that our initial description, which stated "The $(i, j)$th entry of $\\mathbf{B}_k$ is non-zero if the $i$th $(k−1)$-simplex is a boundary simplex of the $j$th $k$-simplex", could have been clearer.
>
> We have clarified in the revised manuscript that the $(i,j)$th entry of $\mathbf{B}_k$ can be either $+1$ or $-1$, depending on the relative orientations of the $i$th $(k−1)$-simplex and the $j$th $k$-simplex. Even though in the introduction we mentioned about orientations of higher-order simplices as “Higher-order simplices are usually oriented, ensuring a consistent node arrangement within each simplex, facilitating tasks like determining information flow directions along the edges.”, we did not formally define the orientations of features on the simplicial complex in relation to the reference node ordering. Therefore, we have now specified in the revised manuscript that signs of features on simplices are determined based on the reference node arrangement within each simplex. Additionally, when features are unoriented, we clarify that there is no need for a reference orientation of simplices. In such cases, we consider unoriented incidence matrices.
>
> To this end, we have incorporated the following points in Section 2 (Background) at appropriate locations:
>
> "Each simplex has an orientation defined by a standardized vertex order, typically ascending or descending, establishing a standard node arrangement within each simplex."
>
> "The non-zero entries of an oriented incidence matrix $\mathbf{B}_k$ can be either $+1$ or $-1$, reflecting the relative orientations of the $i$th $(k−1)$-simplex and the $j$th $k$-simplex."
>
> "The sign of feature $\mathbf{X}_k[\sigma_k]$ is determined based on the reference orientation of $\sigma_k$."
>
> "For unoriented features, there is no need for a reference orientation of simplices. In such cases, we consider unoriented incidence matrices. Specifically, the $(i, j)$th entry of an unoriented incidence matrix  $\mathbf{B}_k$ is $1$ if the $i$th $(k−1)$-simplex is a boundary simplex of the $j$th $k$-simplex, and $0$ otherwise."
>
> We believe that these revisions will enhance the comprehension of the orientation equivariance concept within our paper.

---

> > ### Author Response · Authors · 2023-11-20
> > **On Validity of the Example in Section 4.1**
> >
> > We appreciate your feedback regarding Section 4.1. In Section 4.1, we consider unoriented simplices. The details are as follows.
> >
> > In Section 4.1, we explore example functions that adhere to the conditions in Theorem 4.2, demonstrating that SaNN is as powerful as the SWL test. It is important to note that the SWL test differentiates non-isomorphic simplicial complexes based on structure, not the features they carry as in the commonly used WL test. Consequently, it assumes uniform initial features (colors) across all simplices, which are then updated iteratively [Please refer to Section 3 in Bodnar et al., 2021].
> >
> > Our goal in showing equivalence to the SWL test is to demonstrate that SaNN assigns distinct embeddings to two non-isomorphic simplicial complexes, irrespective of their features, if the SWL test assigns different colors to their structures. In this context, we consider unoriented uniform features across all simplices, thus dealing with unoriented simplicial complexes.
> >
> > However, SaNN is also capable of handling oriented features in practice. For instance, in trajectory prediction where flow has orientation, we consider oriented simplicial complexes. In such scenarios, we can also establish orientation equivariance as detailed in the paper, hence validating Property 1. In sum, the two settings are different as highlighted by the reviewer and we have clarified this in the paper. To enhance clarity, we have slightly revised the introductory text of Section 4.1 as follows:
> >
> > "In this section, we discuss example functions that fulfill the conditions outlined in Theorem 4 and demonstrate that SaNN is as powerful as the SWL test. The SWL test [cf. Appendix C] distinguishes non-isomorphic simplicial complexes based on structure, assuming uniform initial features (colors) across all simplices. To establish equivalence with the SWL test, we consider simplicial complexes with a uniform scalar feature $a$ on all simplices, without attributing any orientation to the features or the simplices. However, it is worth noting that SaNN can also process oriented features in practice, and in such cases, we work with oriented simplicial complexes (or incidence matrices) as defined in Section 2."

---

> > > ### Author Response · Authors · 2023-11-20
> > > **Clarifying SaNN's Equivalence to the SWL Test**
> > >
> > > In response to your comment, we reemphasize that isomorphism testing primarily focuses on the structure of simplicial complexes, not the features they carry. The objective is that SaNN should assign distinct embeddings to two different structures that the SWL test identifies as non-isomorphic. When establishing equivalence to the SWL test, our reference to requiring injective functions for aggregation pertains to structure, not initial features. This means that we need distinct embeddings for two unique structures, assuming all features are identical. As we theoretically prove in the paper, under the conditions stated in Theorem 4.2, starting with uniform initial features (colors) across all simplices, SaNN is as powerful as the SWL test in distinguishing the structures of non-isomorphic simplicial complexes.

---

> > > > ### Author Response · Authors · 2023-11-20
> > > > **Correcting the Minor Error in the Insights Section**
> > > >
> > > > Thank you for pointing it out. Indeed, you are correct. Orientation equivariance does not automatically equate to superior performance. Similarly, good performance on a specific dataset does not necessarily imply orientation equivariance. Our intention was to highlight that SaNN's good performance signifies its effective use of the orientations of flows for trajectory prediction. Therefore, we will revise the sentence as follows:
> > > >
> > > > "The good performance of SaNN also signifies its effective use of the orientations of flows for trajectory prediction."

---

> ### Author Response · Authors · 2023-11-20
> **Clarification on Orientations of Simplices in Experiments.**
>
> In our implementation of SaNN, we use oriented incidence matrices for feature aggregation when the features are oriented. On the other hand, we use unoriented incidence matrices when the features are unoriented.
>
> In the numerical experiments for trajectory prediction, the input features are oriented flows on simplicial complexes, prompting us to use oriented incidence matrices for aggregation. For other experiments such as graph classification and simplicial closure prediction, no input features were given on the simplicial complexes. We constructed initial features on simplices as the sum of upper and lower adjacent simplices. As these features are unoriented, we utilized unoriented incidence matrices for feature aggregation.
>
> To further clarify, we have included the following text in the respective subsections of Section 5 in the revised manuscript:
>
> 1. In the Trajectory Prediction section: "Trajectory prediction involves predicting the next node in a sequence formed by a series of nodes connected by edges, with oriented flows on the edges. As the features are oriented, we use oriented incidence matrices for aggregation."
>
> 2. In the Simplicial-Closure Prediction section: "In all simplicial closure prediction experiments, we consider the initial features on simplices as the cumulative count of their lower and upper adjacent simplices. Given that these are unoriented, we use unoriented incidence matrices for aggregation."
>
> 3. In the Graph Classification section: "For all graph classification experiments, we consider the initial features on simplices as the cumulative count of their lower and upper adjacent simplices.  As these are unoriented, we use unoriented incidence matrices for aggregation."

---

> > ### Author Response · Authors · 2023-11-20
> > **On the Expressiveness and Simplicial Awarenessof the existing SNNs**
> >
> > We appreciate your feedback and agree that the lack of formal definitions for oriented features and missing to mention the use of unoriented incidence matrices for unoriented features might have caused confusion. We have now clarified these points and expect this to improve the understanding of the theorems and properties related to orientations.
> >
> > Regarding simplicial awareness, this property requires a model's output to be dependent on all incidence operators. All baseline methods in our paper rely on the powers of lower-adjacent matrices, upper-adjacent matrices, or Hodge Laplacian matrices. As none of these powers approach zero, all baselines inherently exhibit simplicial awareness. However, excluding MPSN, the other baselines do not match the expressivity of the SWL test, as they do not account for feature aggregation from boundary and coboundary simplices, while the SWL test does. Furthermore, since the baselines other than MPSN do not consider aggregation from coboundary simplices, the node embeddings from these models cannot be shown to be more expressive than those from the WL test.
> >
> > We will supplement Appendix A with the following text for further clarification:
> > "All baseline methods, including \\texttt{MPSN}, \\texttt{SCNN}, \\texttt{SCoNe}, and \\texttt{S2CNN}, exhibit simplicial awareness as they utilize the powers of lower-adjacent matrices, upper-adjacent matrices, or Hodge Laplacian matrices, none of which approach zero. However, except for \\texttt{MPSN}, these baselines do not match the SWL test in terms of expressivity, as they do not account for feature aggregation from boundary and coboundary simplices, which is considered by the SWL test. Furthermore, since the baselines other than MPSN do not consider aggregation from coboundary simplices, the node embeddings from these models cannot be shown to be more expressive than those from the WL test. We summarize the properties of \\texttt{SaNN} and the existing SNN models as follows:
> >
> > | Method | Aggregations | More expressive than WL | As expressive as SWL | Simplicial-awareness |
> > |--------|--------------|------------------------|----------------------|----------------------|
> > | SaNN   | Precomputed  | Yes                    | Yes                  | Yes                  |
> > | MPSN   | Sequential   | Yes                    | Yes                  | Yes                  |
> > | SCoNe  | Sequential   | No                     | No                   | Yes                  |
> > | SCNN   | Sequential   | No                     | No                   | Yes                  |

---

> > > ### Author Response · Authors · 2023-11-20
> > > **Addressing Feedback and Revisions Made**
> > >
> > > We appreciate your insightful feedback regarding the clarity of orientations in our work. In our revised manuscript, we have provided comprehensive clarifications on this aspect. We believe that these modifications, along with the substantial results presented in the paper, strengthen the overall quality of the manuscript. We hope you find these revisions address your concerns.

---

### Official Review · Reviewer_LALK · 2023-10-30

**Soundness:** 4 excellent
**Presentation:** 3 good
**Contribution:** 2 fair
**Rating:** 6
**Confidence:** 3

**Summary:**

The authors present a Simplicial Graph Neural Network, which considers higher-order structures in the input graphs. In comparison to previous work, the features from k-simplices are precomputed without trainable parameters and only then fed into a GNN. This leads to lower runtime during training since features can be reused in each epoch, which is validated by the authors theoretically and empirically.

The authors prove that their method is more powerful than the WL test and as powerful as the Simplicial WL (SWL) test, when it comes to distinguishing non-isomorphic subgraphs. Further, they prove permutation equivariance, orientation equivariance, and simplicial-awareness.

The method is evaluated on trajectory prediction, simplicial closure prediction, and graph classification, where it is on par/slightly outperforms previous works with better training runtimes.

**Strengths:**

- The goal of the work, achieving better scalability of expressive networks by using non-parametric simplicial encoders makes sense.
- The authors thoroughly analyze their method theoretically and provide proofs for all relevant properties.
- The presented method seems to find a good trade-off between expressiveness, runtime and empirical quality.
- There is theoretical value in the non-parametric encoder for simplices that keeps equivariant properties and simplicial-awareness
- The paper is mostly well written

**Weaknesses:**

- Runtime and asymptotic comparisons in this work are done by excluding the precomputation of features. I think this is misleading, since in practice, the precomputation is certainly part of the computation, especially during inference. Thus, the presented gains seem to be only valid during training, when the features need to be computed only once for many iterations of training.
- At the same time, the method only performs on par with previous work, with small gains on some datasets.
- The method requires many hyper parameter choices like hops, T, k, which seem to have different optimal settings on different datasets. The result quality differs substantially depending on the configuration.
- I am skeptical regarding the practical relevance of the presented method due to above reasons.

- The method lacks conceptual novelty. The main idea of precomputing features by non-learnable functions has been seen in other areas, e.g. non-parametric GNNs. The general structure of the work follows a long line of work about GNN expressiveness (higher order and WL-level) without presenting many novel insights.

**Questions:**

- I wonder how the method compares to previous methods in inference runtime, when feature precomputation needs to be included.

-----------
I thank the authors for proving the precomputation times explicitly and for replying to other concerns I have - this is certainly helpful to evaluate the differences to previous work.

In general, I am still on the fence and still doubt the significance of the contribution. However, I acknowledge that other reviewers find value in it and, thus, slightly raise my score. I am not opposing this paper to be accepted, as I think it is a thorough and well executed work.

---

> ### Author Response · Authors · 2023-11-20
>
> We sincerely appreciate your positive feedback on our work's objectives, theoretical analysis, trade-off balance, the value of our non-parametric encoder, and the overall quality of our writing.

---

> ### Author Response · Authors · 2023-11-20
> **Addressing the Inclusion of Precomputation Times in Runtime Comparisons**
>
> We appreciate your feedback regarding the inclusion of precomputation times in our overall computation assessment. While we agree that precomputation is indeed part of the overall process, it generally constitutes a relatively insignificant portion compared to the cumulative time spent on training and testing.
>
> It is important to note that our method requires precomputation of features just once before the training begins. On the other hand, conventional SNNs require this computation for every layer, where the number of layers correspond to the neighborhood depth from which features are aggregated, and for each epoch and across all orders of simplices. In Appendix B, we do provide a theoretical comparison of overall computational complexities of SaNN and the existing SNN models. The overall time complexity of the existing SNN models is about $\mathcal{O}\left(T\left(\left(2 N_k^2 D_k+N_k N_{k-1} D_{k-1}+N_k N_{k+1} D_{k+1}\right)+\left(3 N_k D_k^2+N_k D_{k-1}^2+N_k D_{k+1}^2\right)\right)\right)$, which is dominated by the first term containing $2 N_k^2 D_k, N_k N_{k-1} D_{k-1}$, and $N_k N_{k+1} D_{k+1}$, corresponding to feature aggregation in every layer. Since feature aggregation happens only once before training as a pecomputation step, an SaNN model capturing information from $0, \ldots, T$-hop neighborhood of $k$-simplices has a time complexity of $\mathcal{O}\left(T\left(3 N_k D_k^2+N_k D_{k-1}^2+N_k D_{k+1}^2\right)\right)$, which is negligibly small when compared to the computational complexity of the existing SNN models.
>
> We acknowledge that the time savings during inference may not be as pronounced as those during training, but we assert that the substantial enhancements during the training phase are quite significant, given that training time is the primary contributor to overall computation time, which is the sum of precomputation time, training time and inference time.
> Nonetheless, in response to your suggestion, we have incorporated precomputation times into our analysis   to provide a more comprehensive view of our method's performance. We present here the precomputation and per epoch training time values of SaNN and the per epoch training time values of baseline SNN models for the three tasks at hand. The first terms in the runtime values of SaNN correspond to the precomputation times and the second terms to the per epoch training times.
>
> Trajectory prediction:
> | Dataset | Ocean | Synthetic | Planar | Mesh |
> |---------|-------|-----------|--------|------|
> | ScoNe   | 0.4   | 3.2       | 30.2   | 18.9 |
> | SCNN    | 1.9   | 4.2       | 36.3   | 29.8 |
> | SaNN    | 0.01,0.1 | 0.06,2.5 | 1.22,26.8 | 0.39,8.1 |
>
>
> Simplicial closure prediction:
> | Dataset      | Enron   | High-school | Primary-school | NDC-classes | Math-sx     |
> |--------------|---------|-------------|----------------|-------------|-------------|
> | MPSN         | 255     | 413         | 3499           | -           | -           |
> | SCNN         | 17      | 401         | 1891           | -           | -           |
> | SaNN         | 0.01, 3 | 0.05, 112   | 0.76, 916      | 0.26, 13    | 95.91, 52883|
>
> Graph classification:
> | Dataset  | Proteins | NCI1 | IMDB-B | Reddit-B | Reddit-M |
> |----------|----------|------|--------|----------|----------|
> | MPSN     | 33       | 292  | 46     | 242      | 1119     |
> | SaNN     | 3.6, 0.4 | 6, 58| 4, 8   | 6, 45    | 34, 104  |
>
>
> The values provided represent the training time per epoch. Out-of-memory results are indicated by −. The total training time for SaNN is calculated as follows:
> Total training time = Precomputation time + (Number of epochs * Training time per epoch). This total is substantially less than the precomputation times. Furthermore, it is also worth noting that for any neural network model, the larger computational burden lies in the training phase rather than the inference phase. In our study, we have demonstrated both theoretically and experimentally that our training times are significantly reduced compared to other models. Therefore, the overall computational efficiency of our method remains superior.

---

> > ### Author Response · Authors · 2023-11-20
> > **Addressing the Comparative Performance of SaNN**
> >
> > It is indeed correct that SaNN demonstrates comparable, and occasionally superior, performance to the baselines on some datasets. However, the real strength of SaNN lies in its significant training time improvements. This means that while maintaining competitive accuracy, SaNN offers substantial computational gains, which is a crucial factor in practical applications. Table 2 illustrates that existing SNN models often encounter out of memory errors when handling very large datasets. In contrast, our proposed SaNN model not only significantly reduces computational time, but also delivers performance that is consistently competitive with, and at times surpasses, that of the existing SNN models.

---

> > > ### Author Response · Authors · 2023-11-20
> > > **Addressing Concerns on Hyperparameter Choices**
> > >
> > > We acknowledge your concern about the numerous hyperparameters in our method, such as depth $(T)$, order of simplicial complex $(K)$, and hops. It is true that optimal settings can vary across different datasets, potentially impacting the quality of results.
> > >
> > > However, it is important to note that hyperparameters like depth $T$ and order $K$ are not unique to our model, but are intrinsic to any model dealing with simplicial complexes. To understand the impact of information from different depths and simplices of varying orders, we conducted an ablation study. Our findings from this study are detailed in Appendix J. We have also summarized key insights from this analysis in Section 5 under "Additional Insights." We believe these observations can guide the selection of hyperparameters for different applications.

---

> > > > ### Author Response · Authors · 2023-11-20
> > > > **Addressing Concerns on Practical Relevance**
> > > >
> > > > Regarding the concern of practical relevance of our method, we want to highlight that one of the main goals of the paper is to have a simpler model of practical value that can be theoretically characterized. Reduced training time not only enables more frequent model retraining, catering to real-time applications, but also offers scalability when dealing with larger datasets. While inference time is indeed crucial, the ability to train models more efficiently is equally significant in real-world machine learning applications. As indicated in Table 2, existing SNN models struggle with training on very large datasets. However, our SaNN model, despite its significantly lower computational time, achieves performance that is competitive with, and occasionally even superior to, existing SNN models. This balance between computational efficiency and performance highlights the practical value of SaNN, particularly in large-scale applications.
> > > >
> > > > In terms of practicality in hyperparameter selection, our ablation study provides valuable insights. We found that local neighborhood information is vital across all tasks, while information from larger hops seems less critical. For simplicial closure prediction, our study revealed that the constituent edges of open triangles carry the most crucial information about whether a simplex will be formed. This agrees with the observation made in Benson et al. (2018), which states that the tie strengths of the edges in an open triangle positively impact its simplicial closure probability. We believe insights such as these offer practical guidance for choosing hyperparameters in different applications.

---

> > > > > ### Author Response · Authors · 2023-11-20
> > > > > **Addressing Concerns about the Novelty of SaNN**
> > > > >
> > > > > Indeed, our work draws inspiration from graph-based methods such as GAMLPs, SPIN, and SIGN, which utilize the idea of precomputing features using non-learnable functions. Nevertheless, the definition of a neighborhood in SaNN is fundamentally different as discussed in the paper.
> > > > >
> > > > > A straightforward extension of the precomputation step in GAMLPs, SPIN, or SIGN to simplicial complexes would involve precomputing the features of simplices using integer powers of the Hodge Laplacian matrix. However, this approach fails to account for the information embedded in the boundary and co-boundary adjacent simplices, which, as we demonstrate in our work, is crucial for proposing an efficient model that matches the power of the SWL test.
> > > > > Our work presents a novel recursive aggregation approach, which, subject to the selection of specific functions for generating embeddings, leads to node embeddings that are more expressive than those produced by traditional GNN models. Moreover, we offer theoretical evidence that SaNN, through our proposed aggregation and transformation methodology, adheres to orientation equivariance and simplicial awareness, which are characteristics specific to simplicial complex-based models. We believe these unique attributes highlight the novelty of our method when compared to conventional graph-based approaches.

---

### Official Review · Reviewer_7LnN · 2023-11-06

**Soundness:** 3 good
**Presentation:** 4 excellent
**Contribution:** 3 good
**Rating:** 8
**Confidence:** 2

**Summary:**

The authors propose a class of simple simplicial neural network models, referred to as simplicial-aware neural
networks (SaNNs), which leverage precomputation of simplicial features. The authors theoretically demonstrate that under certain conditions, SaNNs are better discriminators of non-isomorphic graphs than the WL and SWL test. Empirically, SaNNs are shown to perform competitively against other SNNs and GNNs on tasks such as trajectory prediction, simplicial closure prediction, and several graph classification tasks over various datasets.

**Strengths:**

1. The theoretical results are intriguing. Indeed, a competitor to the WL and SWL tests would be a valuable contribution to the graph ML community.

2. A wide variety of benchmarks over several tasks and datasets are conducted to demonstrate the efficacy and efficiency of SaNNs.

3. SaNNs inherit several valuable invariance properties of other SNNs including permutation invariance, orientation invariance, and simplicial-awareness.

4. Compared to MPSNs, consideration of higher-order simplices does not blow up computation complexity.

**Weaknesses:**

1. It is unclear for a research with limited expertise in this rather niche area to conclude the strength of the conditions prescribed in Theorems 4.1 and 4.2. (See questions.)

2. There do not appear to be any results describing the pre-computation time which should be included in any run-time comparisons which I imagine should scale near-exponentially with graph size and order of simplices considered.

3. SaNNs are often not outright the winner in terms of prediction accuracies for the tasks displayed in Tables 1 and 3. For example, in Table 1, the SaNN is outcompeted by Projection and Scone on 3/4 of the datasets and the run-time savings of SaNN are not significant enough to justify usage of the SaNN. In Table 3, SaNN is not the leader in 4/5 of the datasets and it is not even the fastest. On the other hand, the time savings against MPSN are quite significant, but since many practitioners of graph learning expect training to take significant amounts of time, accuracy is the topmost priority, so there wouldn't be a strong enough justification to go with a SaNN.

**Questions:**

1. Is assuming the learnable transformation functions $g_k^{(t)}\cdot)$ are injective too strong? Although the MLPs will be injective, appealing to the Universal Approximation Theorem to declare that $g_k$ can be injectively-approximated is probably not practical.

2. I may have missed this but are pre-computation times explicitly indicated in the results?

---

> ### Author Response · Authors · 2023-11-20
>
> We sincerely appreciate your positive feedback and recognition of the potential impact of our work in the graph ML community. Your acknowledgment of the strengths of our SaNN model, its invariance properties, and its computational efficiency is highly encouraging. Thank you.

---

> > ### Author Response · Authors · 2023-11-20
> > **Clarification Regarding the Usage of the Universal Approximation Theorem**
> >
> > We agree that citing the Universal Approximation Theorem to propose that $g_k$ can be injectively approximated may seem like a strong assumption. This assumption is primarily used for theoretical justification, and we understand its potential limitations in practical applications.
> > However, it is important to note that we have only used multilayer perceptrons (MLPs) as one possible example of the many functions that $g_k$ could be. It is possible to choose any suitable learnable injective function. Our choice of MLPs is simply an example, and it has indeed performed well in our numerical experiments. Therefore, while the injectivity assumption offers theoretical support, it does not diminish the practical effectiveness of our proposed method.

---

> ### Author Response · Authors · 2023-11-20
> **Regarding Inclusion of Precomputation Times**
>
> We appreciate your feedback regarding the inclusion of precomputation times in our overall computation assessment. While we agree that precomputation is indeed part of the overall process, it generally constitutes a relatively insignificant portion compared to the cumulative time spent on training and testing.
> Nonetheless, in response to your suggestion, we incorporated precomputation times into our analysis to provide a more comprehensive view of our method's performance. We present here the precomputation and per epoch training time values of SaNN and the per epoch training time values of baseline SNN modesl for the three tasks at hand. The first terms in the runtime values of SaNN correspond to the precomputation times and the second terms to the per epoch training times.
> Trajectory prediction:
> | Dataset | Ocean | Synthetic | Planar | Mesh |
> |---------|-------|-----------|--------|------|
> | ScoNe   | 0.4   | 3.2       | 30.2   | 18.9 |
> | SCNN    | 1.9   | 4.2       | 36.3   | 29.8 |
> | SaNN    | 0.01,0.1 | 0.06,2.5 | 1.22,26.8 | 0.39,8.1 |
>
> Simplicial closure prediction:
> | Dataset      | Enron   | High-school | Primary-school | NDC-classes | Math-sx     |
> |--------------|---------|-------------|----------------|-------------|-------------|
> | MPSN         | 255     | 413         | 3499           | -           | -           |
> | SCNN         | 17      | 401         | 1891           | -           | -           |
> | SaNN         | 0.01, 3 | 0.05, 112   | 0.76, 916      | 0.26, 13    | 95.91, 52883|
>
> Graph classification:
> | Dataset  | Proteins | NCI1 | IMDB-B | Reddit-B | Reddit-M |
> |----------|----------|------|--------|----------|----------|
> | MPSN     | 33       | 292  | 46     | 242      | 1119     |
> | SaNN     | 3.6, 0.4 | 6, 58| 4, 8   | 6, 45    | 34, 104  |
>
> The values provided represent the training time per epoch. Out-of-memory results are indicated by −. The total training time for SaNN is calculated as follows:
> Total training time = Precomputation time + (Number of epochs * Training time per epoch). This total is substantially less than the precomputation times.

---

> > ### Author Response · Authors · 2023-11-20
> > **Addressing the Comparative Performance of SaNN**
> >
> > We agree that SaNN does not consistently outperform other models in terms of prediction accuracy for certain datasets. However, the true strength of SaNN lies in its substantial runtime improvements and theoretical characterization while maintaining competitive accuracy. This computational efficiency is crucial for practical applications.
> > The reduced training time that SaNN offers enables more frequent model retraining, which is particularly beneficial for real-time applications. It also ensures scalability for larger datasets. As shown in Table 2, traditional SNN models struggle with large datasets, while SaNN excels, especially with datasets in simplicial closure prediction where existing SNNs face memory constraints.
> >
> > In Table 3, although SaNN is not the leader in average accuracies for most datasets, the high standard deviations suggest significant overlap in accuracy results, rendering the differences between the top-performing models statistically insignificant. While SPIN is the fastest for graph classification, it is important to note that it is not applicable to applications involving simplicial complexes such as simplicial closure prediction and trajectory prediction.
> >
> > While accuracy is indeed a priority in many applications, the balance between accuracy and computational efficiency is an equally critical factor in real-world scenarios. SaNN offers a valuable trade-off in this context, making it a viable option, particularly for large-scale applications.

---

> > > ### Comment · Reviewer_7LnN · 2023-11-21
> > > **Thanks for the response**
> > >
> > > I'd like to thank the authors for addressing my questions and providing pre-computation times which are indeed not significant. I am maintaining my score.

---

### Official Review · Reviewer_jAzK · 2023-11-08

**Soundness:** 4 excellent
**Presentation:** 3 good
**Contribution:** 4 excellent
**Rating:** 8
**Confidence:** 4

**Summary:**

The paper describes an efficient, and effective approach for learning representations for simplices in a simplicial complex. The central idea is that of using injective functions for aggregating simplicial features, as it ensures that the embeddings are unique. The simplicial features are aggregated over upper, lower, boundary and co-boundary adjacencies. The paper provides precise definitions and theorems and statements on the properties of the networks. The proofs are summarized in the main body and provided in full detail in the appendices. The method is further experimentally validated and shows that the proposed model (SaNN) is both efficients (significantly faster than any of the other baselines) and effective (performance within the uncertainty intervals on accurcies, or above the baselines).

**Strengths:**

1. I am impressed by the clarity of presentation in the paper. I find talking and reading about simplicial complex often a messy business given all the types of simplices and adjacencies, and the abstract notion in the first place. It is clear that the authors though well about how to present the math. This includes proper use of figures.
2. The goal of the paper itself -efficiency whilst not compromising on expressivity- is relevant and important, and it is great to see the authors succeeding in reaching this goal.
3. I appreciate the summary of the proofs after the formal statements.
4. Next to a sound theoretical exposition, the experiments are thorough as well and include many ablation studies that are used to distill insightful take home messages.

**Weaknesses:**

I only have 1 important concern:

1. Although the main principles are clear, I am still confused about the actual architecture/predictive models. In the end we have equation 8, but it describes a representation for each of the $N$ sets of $k$-simplices, each consisting of the $N_k$ simplices. It is unclear how to distill a global prediction out of all these representations, as would be needed for e.g. the classification tasks. Details on how the architectural design for each of the benchmarks is missing.

**Questions:**

Could you respond to the above concern, and additionally address the following questions/comments?

2. On several occasions the notion of "non-deep baselines" is used. What is meant by this. Could you clarify what non-deep means here, which methods are these?

3. In section 2 when presenting the symbols it is mentioned that $k=1,2,\dots,N+1$. Does $k$ always run up all the way to $N+1$?

4. In section 4. The sentence that starts with "The theorem implies that any arbitrary ..." is extremely long and hard to comprehend. I suggest to split it 2 or 3 sentence to improve readability.

5. Just above property 1 it is mentioned "other commonly used sum, mean, or max read-out functions are not injective" I am not fully sure I understand it correctly. The paragraph above explains that sum aggregation is the best injective aggregator, in contrast to mean aggregation. I think the statement that I just quoted is about aggregating over the different $\mathbf{Y}$'s? Perhaps this can be clarified.

6. In the tables: since colors red and blue are used you might as well color the text in the caption as well. I.e. "The {\color{red}first} and {\color{blue}second} best performances ..."

7. The insights section says "The deep models are observed to perform exceptionally better than logistic regression", where do I see this? Logistic regression taking what as input? Could this be clarified.

Thank you for considering my comments and questions.

---

> ### Author Response · Authors · 2023-11-20
>
> We deeply appreciate your kind words regarding the clarity of our presentation and the balance between efficiency and expressivity in our work. Thank you for acknowledging the thoroughness of our theoretical and experimental sections.

---

> > ### Author Response · Authors · 2023-11-20
> > **Clarification on Architectures/Predictive Models**
> >
> > The predictive models, which are used to learn task-specific embeddings from the embeddings of all simplices (as calculated using Equation 8), are detailed in Appendix H.
> >
> > To use the edge embeddings learnt using SaNN to perform trajectory prediction, we map the embeddings from the vector space of edge embeddings to the vector space of node embeddings by multiplying them by the incidence matrix $\mathbf{B}_1^T$. We consider the node embeddings of only those nodes that have a possibility of being the next in the trajectory, which are the neighboring nodes of the last node in a trajectory. For a trajectory t, which ends at node i, we use a masking function to extract the embeddings corresponding to the neighboring nodes of the node i. We do this for node embeddings from different depths and concatenate them. We then give the combined node embeddings as input to a decoder MLP which maps the node embeddings to a vector space of dimension same as the maximum number of neighboring nodes of the last node in any trajectory in the dataset. The last layer of the decoder is a softmax activation function, the output of which for a trajectory is a vector containing the probabilities with which the neighboring nodes of the trajectory’s last node become the successor node of that trajectory. For all the datasets, we train SaNN in a supervised manner using the cross-entropy loss function.
> >
> > To perform simplicial closure prediction for triangles, we concatenate the embeddings of the constituent nodes and edges in the open triangles in the training set. The concatenated embeddings are given as input to a decoder MLP with a sigmoid activation function in the last layer, the output of which is the probability of the open triangles forming a simplex. For all the datasets, we train SaNN in a supervised manner using the binary cross-entropy loss function. Since the datasets are heavily skewed with many negative examples (open triangles that never get closed) than positive examples (open triangles that get closed), we perform oversampling of the positive examples in each training batch and use a weight for positive examples in the binary cross-entropy loss function.
> >
> > To perform graph classification using the embeddings learnt using SaNN, we combine embeddings of the constituent simplices of all orders in the clique-lifted graphs. Since the number of k-simplices in a training example (a clique-lifted graph) is large concatenating embeddings of all the k-simplices in a training example will result in very high-dimensional embeddings. Therefore, we use summation as the readout function at $k$-simplices level. We, finally, use concatenation as the global-level readout function. Specifically, we take a summation of all the node embeddings, edge embeddings, and triangle embeddings in a graph and concatenate the nodelevel, edge-level, and triangle-level embeddings of the simplicial complex to get one global representation for the simplicial complex. The global embeddings of graphs are given as input to a decoder MLP with either a sigmoid activation function or a softmax activation function as the final layer depending on whether the dataset has two or more classes. For all the datasets, we train SaNN in a supervised manner using the cross-entropy loss function, which takes the output of the decoder and the true graph labels as the input.
> >
> > The hyperparameters used in SaNN to implement the different tasks are summarized in Tables 5, 7, and 9. We have maintained consistency in our experiments by using the same hyperparameters, such as depth ($T$), order ($K$), and hidden dimensions, across all baseline SNN models. This was done to ensure a fair computational comparison. The predictive models for the three tasks are also the same for all the neural network-based models. Due to space constraints, we provided all these details in the Appendix.
> >
> > To provide further clarity about the hyperparameters used for benchmarks, we have added the following line to Appendix H:
> > "For a fair computational comparison, the same predictive models, and hyperparameters, namely depth ($T$), order ($K$), and hidden dimensions, were utilized for all the baseline SNN models.

---

> > > ### Author Response · Authors · 2023-11-20
> > > **Clarification Regarding 'Non-Deep Baselines'**
> > >
> > > In our paper, the term "non-deep" refers to models that do not utilize deep learning methodologies. Examples include techniques like Projection for trajectory prediction and Logistic Regression for simplicial closure prediction.
> > >
> > > The projection based method is used as a baseline for trajectory prediction in Roddenberry et al. (2021). In the projection-based method for trajectory prediction, a mapping that projects the input edge signal corresponding to a trajectory onto the kernel of the Hodge Laplacian is considered. The output is passed through a softmax layer to pick a successor node in the trajectory.
> > >
> > > For simplicial closure prediction, we use logistic regression as a non-deep baseline, following the approach proposed by Benson et al. (2018). In this context, logistic regression takes as its input the harmonic, geometric, and arithmetic means of the three edge weights in the open triangle. This is a method suggested by Benson et al. (2018) in their work.

---

> > > > ### Author Response · Authors · 2023-11-20
> > > > **Clarification Regarding the Order of Simplicial Complexes**
> > > >
> > > > Thank you for bringing this to our attention. You are correct; the variable $k$ does not necessarily run up all the way to $N+1$. This was an oversight on our part. In the revised manuscript, we introduced the notation of the order of the simplicial complex as $K$ and corrected the upper limit of $k$ wherever applicable.

---

> > > > > ### Author Response · Authors · 2023-11-20
> > > > > **Revising the Lengthy Sentence**
> > > > >
> > > > > Thank you for your feedback. You are correct that the sentence in Section 4 was quite lengthy. To improve clarity, we haven broken it down as follows and have made changes in the manuscript:
> > > > >
> > > > > “The theorem we present implies that any arbitrary extension of GAMLPs (Chen et al., 2020), SPIN (Doshi & Chepuri, 2022), or SIGN (Rossi et al., 2020) to higher-order simplices does not result in the node-embeddings from SaNN having a superior expressive power GNNs. In one possible extension, we could replace the integer powers of adjacency matrices in these graph models with those of the Hodge Laplacian matrices. These matrices generalize the graph Laplacian to simplicial complexes and are defined as the sum of upper and lower Laplacians. However, even with this modification, the expressive power of these extended models does not surpass that of GNNs.”

---

> > > > > > ### Author Response · Authors · 2023-11-20
> > > > > > **Clarification Regarding Injectivity of Sum Aggregator and Readout Functions**
> > > > > >
> > > > > > When we discuss the injectivity of an aggregation function, we are referring to its ability to distinguish between non-isomorphic simplicial complex structures with identical initial features. In this context, the sum aggregation function is effective at producing distinct outputs for distinct structures after aggregation.
> > > > > >
> > > > > > The sentence above Property 1, however, is discussing the read-out function which combines features of simplices aggregated from different types of neighborhoods. Here, the sum function may not be injective, as combining different sets of embeddings through summation could result in identical outputs.

---

> > > > > > > ### Author Response · Authors · 2023-11-20
> > > > > > > **Color Coding in Captions**
> > > > > > >
> > > > > > > Thank you for your suggestion regarding the color coding in the captions of tables. We have incorporated this change in our revised manuscript.

---

> > > > > > > > ### Author Response · Authors · 2023-11-20
> > > > > > > > **Clarification Regarding Logistic Regression Baseline**
> > > > > > > >
> > > > > > > > In numerical experiments in the paper, we use logistic regression as a non-deep baseline for simplicial closure prediction, following the approach proposed by Benson et al. (2018). In this context, logistic regression takes as its input the harmonic, geometric, and arithmetic means of the three edge weights in the open triangle. This is a method suggested by Benson et al. (2018) in their work.

---

> > ### Comment · Reviewer_jAzK · 2023-11-20
> > **Thank you for the clarification**
> >
> > I checked appendix H which indeed gives sufficient details on the experiments. I also appreciate the given responses to my questions. Although the mentioned minor changes to the paper should be easy to incorporate,  I would appreciate seeing an updated pdf on openreview. Is it possible to do this?
> >
> > Thank you!

---

> > > ### Author Response · Authors · 2023-11-21
> > >
> > > Thank you for your feedback and for acknowledging the details provided in Appendix H. We are glad to hear that the responses to your questions were satisfactory.
> > >
> > > We have uploaded the revised manuscript. The revised version incorporates all the changes that we discussed.
> > >
> > > We appreciate your time and constructive comments throughout this process. Please feel free to review the updated manuscript and share any further thoughts or questions you might have.

---

> > > > ### Comment · Reviewer_jAzK · 2023-11-22
> > > >
> > > > Thank you, much appreciated. I maintain my score.

---

### Author Response · Authors · 2023-11-20
**Revised Manuscript Incorporating Reviewers' Feedback Uploaded**

Dear Reviewers,

Thank you for your time and the positive evaluation of our manuscript. We have now uploaded the revised version, in which we believe we addressed all your concerns. In particular, in the revised version, we (a) defined oriented incidence matrices and clarified when we use oriented and unoriented simplices to improve the exposition; (b) reported precomputation times; (c) introduced the notation of the order of simplicial complexes as $K$ and corrected the upper limit of $k$ where necessary. Please refer to individual comments for responses to your questions and weaknesses.

Best regards,
Authors

---

### Public Comment · ~Rongqin_Chen1 · 2024-11-04
**Request full source code**

Dear Authors,

Thank you for your excellent paper. I am very interested in the proposed method and would like to explore the implementation details further. However, I noticed that the source code provided in the supplementary material appears to be incomplete.

Could you please provide the complete source code? This would greatly assist me in understanding and replicating your results.

Thank you for your time and assistance.

Best regards.

---

### Meta-Review · Area_Chair_Aj5i · 2023-12-10

**Metareview:**

The paper presents an approach for more efficient simplicial complex learning by trading the high computational complexity of high order graph learning (due to the high number of simplicies) with cheaper, precomputed features (computed one-time). The paper provides theory justifying the expressive power achieved when incorporating these features within the WL theoretical framework. Experiments support the efficiency claims compared to existing simplicial complex learning baselines. While reviewers appreciate the method and the computational saving the fact that GNN works have followed this path of adding features and proving WL expressive power reduces a bit the novelty in this work.

**Justification For Why Not Higher Score:**

As mentioned above, while this is a well executed work, the conceptual novelty is somewhat limited.

**Justification For Why Not Lower Score:**

The work does state a useful goal of making higher order graph learning more scalable, which is a worthy and important goal. The paper provides algorithm and theoretical analysis to achieve this goal. There is also an experimental evidence supporting the paper's claim.

---

### Decision · Program_Chairs · 2024-01-16

Accept (spotlight)